# Conformal Prediction using Conditional Histograms

**Matteo Sesia**
Department of Data Sciences and Operations
University of Southern California, USA
sesia@marshall.usc.edu

**Yaniv Romano**
Departments of Electrical and Computer Engineering
and of Computer Science
Technion, Israel
yromano@technion.ac.il

## Abstract

This paper develops a conformal method to compute prediction intervals for non-parametric regression that can automatically adapt to skewed data. Leveraging black-box machine learning algorithms to estimate the conditional distribution of the outcome using histograms, it translates their output into the shortest prediction intervals with approximate conditional coverage. The resulting prediction intervals provably have marginal coverage in finite samples, while asymptotically achieving conditional coverage and optimal length if the black-box model is consistent. Numerical experiments with simulated and real data demonstrate improved performance compared to state-of-the-art alternatives, including conformalized quantile regression and other distributional conformal prediction approaches.

## 1 Introduction

### 1.1 Problem statement and motivation

We consider the problem of predicting *with confidence* a response variable $Y \in \mathbb{R}$ given $p$ features $X \in \mathbb{R}^p$ for a test point $n+1$, utilizing $n$ pairs of observations $\{(X^{(i)}, Y^{(i)})\}_{i=1}^n$ drawn *exchangeably* (e.g., i.i.d.) from some unknown distribution, and leveraging any machine-learning algorithm. Precisely, $\forall \alpha \in (0, 1)$, we seek a prediction *interval* $\hat{C}_{n,\alpha}(X_{n+1}) \subset \mathbb{R}$ for $Y_{n+1}$ satisfying the following three criteria. First, $\hat{C}_{n,\alpha}$ should have finite-sample marginal coverage at level $1 - \alpha$,

$$\mathbb{P}\left[Y_{n+1} \in \hat{C}_{n,\alpha}(X_{n+1})\right] \geq 1 - \alpha. \tag{1}$$

Second, $\hat{C}_{n,\alpha}$ should *approximately* have conditional coverage at level $1 - \alpha$,

$$\mathbb{P}\left[Y_{n+1} \in \hat{C}_{n,\alpha}(x) \mid X_{n+1} = x\right] \geq 1 - \alpha, \qquad \forall x \in \mathbb{R}^p, \tag{2}$$

meaning it should approximate this objective in practice, and ideally achieve it asymptotically under suitable conditions in the limit of large sample sizes. Third, $\hat{C}_{n,\alpha}$ should be as narrow as possible.

We tackle this challenge with conformal inference [24, 35], which allows one to convert the output of any black-box machine learning algorithm into prediction intervals with provable marginal coverage (1). The key idea of this framework is to compute a *conformity score* for each observation, measuring the discrepancy, according to some metric, between the true value of $Y$ and that predicted

by the black-box model. The model fitted on the training data is then applied to hold-out calibration samples, producing a collection of conformity scores. As all data points are exchangeable, the empirical distribution of the calibration scores can be leveraged to make predictive inferences about the conformity score of a new test point. Finally, inverting the function defining the conformity scores yields a prediction set for the test $Y$. This framework can accommodate almost any choice of conformity scores, and in fact many different implementations have already been proposed to address our problem. However, it remains unclear how to implement a concrete method from this broad family that can lead to the most informative possible prediction intervals. Our contribution here is to develop a practical solution, following the three criteria defined above, that performs better compared to existing alternatives and is asymptotically optimal under certain assumptions.

It is worth emphasizing that constructing a short prediction interval with guaranteed coverage is a reasonable approach to quantify and communicate predictive uncertainty in regression problems, although it is of course not the only one. To name an alternative, one could compute a non-convex prediction set with analogous coverage [20], which might be more appropriate in some situations, but is also more easily confusing. For example, it could be informative for a physician to know that the future blood pressure of a patient with certain characteristics is predicted to be within the range [120,129] mmHg. However, it would not be more helpful to report instead the following non-convex region: $[120, 120.012] \cup [120.015, 120.05] \cup [121, 122.7] \cup [123.1, 127.2] \cup [127.8, 129]$ mmHg. Indeed, in the second case it would not be clear (a) whether the multi-modal nature of that prediction is significant or a spurious consequence of overfitting, and (b) how the physician would act upon that prediction any differently than if it had been [120,129] mmHg. Therefore, we focus on prediction intervals in this paper because they are generally easier to interpret than arbitrary regions, and they are also less likely to convey a false sense of confidence.

## 1.2 Preview of conformal histogram regression

Imagine an *oracle* with access to $P_{Y|X}$, the distribution of $Y$ conditional on $X$, which leverages such information to construct optimal prediction intervals as follows. For simplicity, suppose $P_{Y|X}$ has a continuous density $f(y \mid x)$ with respect to the Lebesgue measure, although this could be relaxed with more involved notation. Then, the oracle interval for $Y_{n+1} \mid X_{n+1} = x$ would be:

$$C_\alpha^{\text{oracle}}(x) = \left[ l_{1-\alpha}^{\text{oracle}}(x), u_{1-\alpha}^{\text{oracle}}(x) \right], \tag{3}$$

where, for any $\tau \in (0, 1]$, $l_\tau^{\text{oracle}}(x)$ and $u_\tau^{\text{oracle}}(x)$ are defined as:

$$[l_\tau^{\text{oracle}}(x), u_\tau^{\text{oracle}}(x)] := \underset{(l,u) \in \mathbb{R}^2 \,:\, l \le u}{\arg\min} \left\{ u - l : \int_l^u f(y \mid x) dy \ge \tau \right\}. \tag{4}$$

This is the shortest interval with conditional coverage (2). If the solution to (4) is not unique (e.g., if $f(\cdot \mid x)$ is piece-wise constant), the oracle picks any solution at random. Of course, this is not a practical method because $f$ is unknown. Therefore, we will approximate (4) by fitting a black-box model on the training data, and then use conformal prediction to construct an interval accounting for any possible estimation errors. Specifically, we replace $f$ in (4) with a histogram approximation, hence why we call our method *conformal histogram regression*, or CHR. The output interval is then

$$\hat{C}_{n,\alpha}(x) = \left[ \hat{l}_{\hat{\tau}}(x), \hat{u}_{\hat{\tau}}(x) \right], \tag{5}$$

where $\hat{l}_{\hat{\tau}}(x)$ and $\hat{u}_{\hat{\tau}}(x)$ approximate the analogous oracle quantities in (4). The value of $\hat{\tau}$ in (5) will be determined by suitable conformity scores evaluated on the hold-out data, and it may be larger than $1 - \alpha$ if the model for $f$ is not very accurate. However, if the fitted histogram is close to the true $P_{Y|X}$, the interval in (5) will resemble that of the oracle (3).

Figure 1 previews an application to toy data, comparing CHR to conformalized quantile regression (CQR) [30]; see Section 4.2 for more details. CHR finds the shortest interval such the corresponding area under the histogram is above $\tau$, for any $\tau \in (0, 1]$, and then calibrates $\tau$ to guarantee marginal coverage above $1 - \alpha$; this extracts more information from the model compared to CQR. For example, CHR adapts automatically to the skewness of $Y \mid X$, returning intervals delimited by the 0%–90% quantiles in this example, which are shorter than the symmetric ones (5%–95%) sought by CQR.

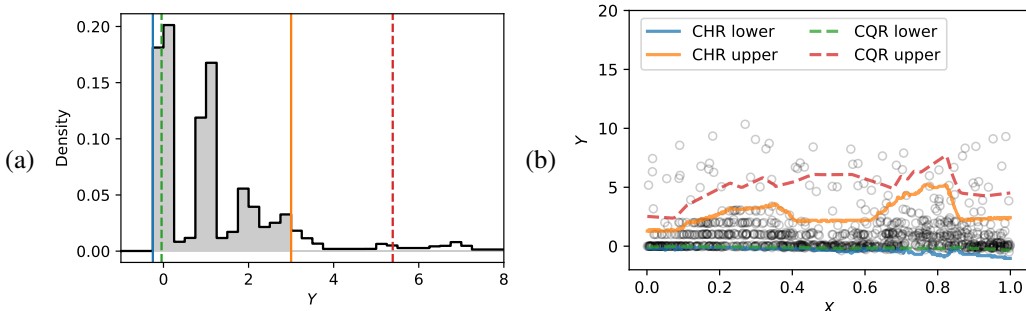

Figure 1: CHR prediction intervals in an example with one variable, compared to those obtained with CQR [30]. Both methods guarantee 90% marginal coverage and are based on the same deep quantile model. (a) Histogram estimate of $P_{Y|X}$ for a point with $X \approx 0.2$. The CHR interval corresponds to the shaded part of the histogram, whose area is approximately 0.9, as marked by the solid vertical lines. The dashed lines denote the CQR interval. (b) Prediction bands for the two methods, as a function of $X$. CHR: empirical marginal coverage 0.9, estimated conditional coverage 0.9, and average length 3.2. The corresponding quantities for CQR are: 0.9, 0.9, and 5.2, respectively.

### 1.3   Related work

This work is inspired by the conformity scores introduced by [31] for multi-class classification, the underlying idea of which can be repurposed here. Nonetheless, the extension to our problem involves several innovations. This paper connects [31] to other conformal methods for continuous responses [24, 25, 35], which sought objectives similar to ours by leveraging quantile regression [17, 23, 30, 32, 36] or non-parametric density estimation [10, 21], sometimes considering multi-modal prediction sets instead of intervals [20]. Our approach also exploits black-box models for distributional estimation; however, we introduce more efficient conformity scores.

We seek the shortest intervals with marginal coverage while approximating as well as possible conditional coverage, although the latter is impossible to guarantee in finite samples [16, 34]. The performances of prior approaches have been measured in terms of these criteria, yet others have not sought them as directly. Indeed, if the black-box model is consistent for $P_{Y|X}$, our method becomes asymptotically equivalent to the oracle (3)–(4), under some technical assumptions. This property does not hold for other existing methods because they tend to produce symmetric intervals, with fixed lower and upper miscoverage rates (the probabilities of the outcome being either below or above the output interval, respectively), which may be sub-optimal if the data have unknown skewness.

## 2   Methods

The proposed method consists of four main components: the estimation and binning of a conditional model for the outcome, the construction of a nested sequence of approximate oracle intervals based on the above, the computation of suitable conformity scores, and their conformal calibration.

### 2.1   Estimating conditional histograms

We partition the domain of $Y$ into $m$ bins $[b_{j-1}, b_j)$, for some sequence $b_0 < \ldots < b_m$. With little loss of generality, assume $Y$ is bounded: $-C = b_0 < Y < b_m = C$, for some $C > 0$. Then, we solve a discrete version of the problem stated in the introduction: we seek the smallest possible contiguous subset of bins with $1 - \alpha$ predictive coverage. If $m$ is large and the bins are narrow, this problem is not very different from the original one, although it is more amenable to solution.

For simplicity, we present our method from a split-conformal perspective [24, 35]; extensions to other hold-out approaches [7, 22, 35] will be intuitive. Let $\mathcal{D}^{\text{train}}, \mathcal{D}^{\text{cal}} \subset \{1, \ldots, n\}$ denote any partition of the data into training and calibration subsets, respectively. $\mathcal{D}^{\text{train}}$ is used to train a black-box model for the conditional probabilities that $Y$ is within any of the above bins: $\forall j \in \{1, \ldots, m\}$,

$$\pi_j(x) := \mathbb{P}[Y \in [b_{j-1}, b_j) \mid X = x]. \tag{6}$$

There exist many tools to approximate $P_{Y|X}$ and obtain estimates $\hat{\pi}_j(x)$ of $\pi_j(x)$, including quantile regression [27, 28, 33], Bayesian additive regression trees [11], or any other non-parametric conditional density estimator [14, 19, 26]. Our method can directly be applied with any of these models, but we found multiple quantile regression to work particularly well [17, 23, 30, 32], and therefore we will focus on it in this paper. Referring to Supplementary Section S1.1 for implementation details and information about the computational cost of the learning algorithm (which is comparable to that required by CQR [30]), we thus take these black-box estimates $\hat{\pi}_j(x)$ as fixed henceforth.

Note that estimating conditional distributions is more challenging if the number of variables is larger. However, this is a fundamental difficulty of high-dimensional regression, not a particular limitation of the proposed CHR. Although our method utilizes conditional histograms learnt from the data, its performance is not directly measured in terms of how closely these resemble the true $P_{Y|X}$. Instead, as we shall see, CHR only needs to detect the possible skewness of $Y \mid X$ and estimate reasonably well some lower and upper quantiles of this conditional distribution. Therefore, its estimation task is not much more difficult than that of CQR [30], as skewness is relatively easy to detect.

## 2.2 Constructing a nested sequence of approximate oracle intervals

For any partition $\mathcal{B} = (b_0, \ldots, b_m)$ of the domain of $Y$, let $\pi = (\pi_1, \ldots, \pi_m)$ be a unit-sum sequence, depending on $x \in \mathbb{R}^p$; this may be seen as a histogram approximation of $P_{Y|X}$ (6). For simplicity, assume all histogram bins have equal width, although this is unnecessary. Then, define the following bi-valued function $\mathcal{S}$ taking as input $x \in \mathbb{R}^p$, $\pi$, $\tau \in (0, 1]$, and two intervals $S^-, S^+ \subseteq \{1, \ldots, m\}$:

$$\mathcal{S}(x, \pi, S^-, S^+, \tau) := \underset{(l,u) \in \{1,\ldots,m\}^2 \, : \, l \leq u}{\arg\min} \left\{ |u - l| : \sum_{j=l}^{u} \pi_j(x) \geq \tau, \, S^- \subseteq [l, u] \subseteq S^+ \right\}. \quad (7)$$

Above, it is implicitly understood we choose the value of $(l, u)$ minimizing $\sum_{j=l}^{u} \pi_j(x)$ among the feasible ones with minimal $|u - l|$, if the optimal solution would not otherwise be unique. Therefore, we can assume without loss of generality the solution to (7) is unique; if that is not the case, we can break the ties at random by adding a little noise to $\pi$. The problem in (7) can be solved at computational cost linear in the number of bins, and it is equivalent to the standard programming challenge of finding the smallest positive subarray whose sum is above a given threshold. Note that we will sometimes refer to intervals on the grid determined by $\mathcal{B}$ as either contiguous subsets of $\{1, \ldots, m\}$ (e.g., $S^-$) or as pairs of lower and upper endpoints (e.g., $[l, u]$).

If $S^- = \emptyset$ and $S^+ = \{1, \ldots, m\}$, the expression in (7) computes the shortest possible interval with total mass above $\tau$ according to $\pi(x)$. Further, if $\pi_j$ is the mass in the $j$-th bin according to the true $P_{Y|X}$, then $\mathcal{S}(x, \pi, \emptyset, \{1, \ldots, m\}, 1 - \alpha)$ is the discretized version of the oracle interval (3)–(4). In general, the optimization in (7) involves the additional *nesting* constraint that the output $\mathcal{S}$ must satisfy $S^- \subseteq \mathcal{S} \subseteq S^+$, which will be needed to guarantee our method has valid marginal coverage [17]. Intuitively, it is helpful to work with a nested sequence because this ensures the prediction intervals are monotone increasing in $\tau$, essentially reducing the calibration problem to that of selecting the appropriate value of $\tau$ that yields the desired marginal coverage. Note that the inequality in (7) involving $\tau$ may not be binding at the optimal solution due to the discrete nature of the optimization problem. However, the above oracle can be easily modified by introducing some suitable randomization in order to obtain valid prediction intervals that are even tighter on average, as explained in Supplementary Section S1.2.

As $\hat{\pi}$ may be an inaccurate estimate of $P_{Y|X}$, we cannot simply plug it into the oracle in (7) and expect valid coverage. However, for any approximate conditional histogram $\hat{\pi}$, we can define a *nested* sequence [17] of (randomized) sub-intervals of $\mathcal{B}$, for different values of $\tau$ ranging from 0 to 1. Then, we calibrate $\tau$ to obtain the desired $1 - \alpha$ marginal coverage. Precisely, consider an increasing scalar sequence $\tau_t = t/T$, for $t \in \{0, \ldots, T\}$ with some $T \in \mathbb{N}$, and define a corresponding growing sequence of subsets $S_t \subseteq \{1, \ldots, m\}$ as follows. First, fix any *starting point* $\bar{t} \in \{0, \ldots, T\}$ and define $S_{\bar{t}}$ by applying (7) without the nesting constraints (with $S^- = \emptyset$ and $S^+ = \{1, \ldots, m\}$):

$$S_{\bar{t}} := \mathcal{S}(x, \pi, \emptyset, \{1, \ldots, m\}, \tau_{\bar{t}}), \quad (8)$$

Note the explicit dependence on $x$ and $\pi$ of the left-hand-side above is omitted for simplicity, although it is important to keep in mind that $S_{\bar{t}}$ does of course depend on these quantities. Figure 2 (second row) visualizes the construction of $S_{\bar{t}}$ in a toy example with $\tau_{\bar{t}} = 0.9$.

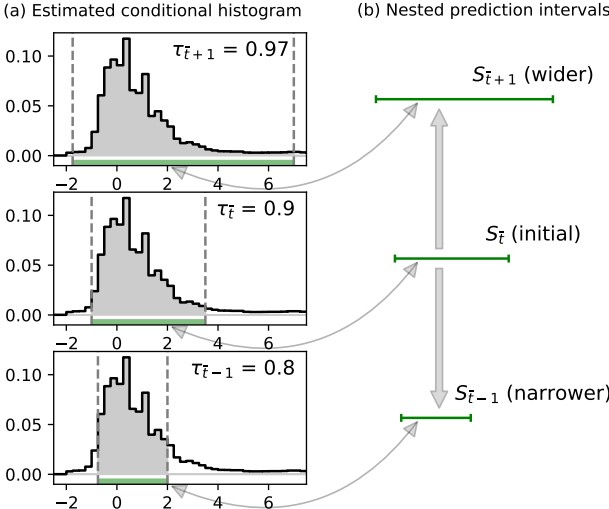

Figure 2: Schematics for the construction of a nested sequence of approximate oracle prediction intervals (8)–(10). (a) Conditional histogram approximation of the distribution of $Y \mid X$, based on a black-box model. The shaded areas delimited by the dashed vertical lines denote the shortest intervals with the desired mass ($\tau$) under the histograms, subject to the nesting constraints. (b) Sequence of prediction intervals. The initial interval $S_{\bar{t}}$ is not subject to any nesting constraints. The wider (above), or narrower (below), intervals must contain $S_{\bar{t}}$ (above), or be contained in it (below).

Having computed the initial interval $S_t$ for $t = \bar{t}$, we recursively extend the definition to the wider intervals indexed by $t = \bar{t} + 1, \ldots, T$ as follows:

$$S_t := \mathcal{S}(x, \pi, S_{t-1}, \{1, \ldots, m\}, \tau_t). \tag{9}$$

See the top row of Figure 2 for a schematic of this step. Similarly, the narrower intervals $S_t$ indexed by $t = \bar{t} - 1, \bar{t} - 2, \ldots 0$ are defined recursively as:

$$S_t := \mathcal{S}(x, \pi, \emptyset, S_{t+1}, \tau_t). \tag{10}$$

See the bottom row of Figure 2 for a schematic of this step. As a result of this construction, the sequence of intervals $\{S_t\}_{t=0}^T$ is nested regardless of the starting point $\bar{t}$ in (8), as previewed in Figure 2. However, different choices of $\bar{t}$ may lead to different sequences, any of which allows us to obtain provable marginal coverage, as discussed next. As our goal is to approximate the oracle in (3)–(4) accurately, the most intuitive choice is to pick $\bar{t}$ such that $\tau_{\bar{t}} \approx 1 - \alpha$. A more involved randomized version of this construction, inspired by the more powerful randomized oracle, is discussed in Supplementary Section S1.2. Note that the randomized version of the nested prediction intervals will be the one applied throughout this paper and, with a slight overload of notation, we will simply refer to it as $\{S_t\}_{t=1}^T$. Note also that we will highlight the dependence of this sequence on $x$ and $\pi$ by writing it as $S_t(x, \pi)$. Further, as we work henceforth with the randomized versions of these prediction intervals (described in Supplementary Section S1.2), we will refer to them as $S_t(x, \varepsilon, \pi)$, where $\varepsilon$ is a uniform random variable in $[0, 1]$, independent of everything else.

### 2.3 Computing conformity scores and calibrating prediction intervals

Given any sequence of nested sets $S_t(x, \varepsilon, \pi)$, we define the following conformity score function $E$:

$$E(x, y, \varepsilon, \pi) := \min \{t \in \{0, \ldots, T\} : y \in S_t(x, \varepsilon, \pi)\}. \tag{11}$$

In words, this computes the smallest index $t$ such that $S_t(x, \varepsilon, \pi)$ contains $y$, as in [17, 31]. Equivalently, one can think of these scores as indicating the smallest value of the nominal coverage $\tau$ in (7) necessary to ensure the observed $Y$ is contained in the prediction interval. Our method evaluates (11) on all calibration samples $(X_i, Y_i)$ using the $\hat{\pi}$ learnt on the training data; for each $i \in \mathcal{D}^{\text{cal}}$, we generate $\varepsilon_i \sim \text{Unif}(0, 1)$ and store

$$E_i = E(X_i, Y_i, \varepsilon_i, \hat{\pi}).$$

Then, we compute prediction intervals for $Y_{n+1}$ by looking at the nested sequence in (8)–(10) corresponding to the new $X_{n+1}$ and selecting the interval indexed by the $1 - \alpha$ quantile (roughly) of $\{E_i\}_{i \in \mathcal{D}^{\text{cal}}}$. The procedure is outlined in Algorithm 1. Note that the only computationally expensive component of this method is the estimation of the conditional histograms (see Supplementary Section S1.1 for details); the construction of the nested prediction intervals and the evaluation of the conformity scores have negligible cost because the optimization problem in (7) is an easy one.

It may be helpful to point out that, if $\pi$ provides an accurate representation of the true conditional distribution of $Y \mid X$, then the above conformity scores are uniformly distributed [31]. In that ideal case, no calibration is needed and indeed our method simply reduces to applying (7) with $\tau = 0.9$ to construct prediction intervals with 90% coverage. In practice, however, $\pi$ can only be a possibly inaccurate estimate of $P_{Y|X}$ (hence why we will refer to it as $\hat{\pi}$ from now on), which means that the distribution of the conformity scores may not be uniform and the conformal calibration is necessary to obtain valid coverage.

The next result states that the output of our method has valid marginal coverage, regardless of the accuracy of $\hat{\pi}$. The proof relies on the sequence $S_t$ being nested; from there, coverage follows from the results of [17, 31]; see Supplementary Section S1.1.

---

**Algorithm 1:** CHR with split-conformal calibration

---

1 **Input:** data $\{(X_i, Y_i)\}_{i=1}^n$, $X_{n+1}$, partition $\mathcal{B}$ of the domain of $Y$ into $m$ equal-sized bins, level $\alpha \in (0, 1)$, resolution $T$ for the conformity scores, starting index $\bar{t}$ for recursive definition of conformity scores, black-box algorithm for estimating conditional distributions.
2 Randomly split the training data into two subsets, $\mathcal{D}^{\text{train}}, \mathcal{D}^{\text{cal}}$.
3 Sample $\varepsilon_i \sim \text{Uniform}(0, 1)$ for all $i \in \{1, \ldots, n + 1\}$, independently of everything else.
4 Using the data in $\mathcal{D}^{\text{train}}$, train any estimate $\hat{\pi}$ of the mass of $Y \mid X$ for each bin in $\mathcal{B}$ (6); see Supplementary Section S1.1 for a concrete approach based on quantile regression.
5 Compute $E_i = E(X_i, Y_i, \varepsilon_i, \hat{\pi})$ for each $i \in \mathcal{D}^{\text{cal}}$, with the function $E$ defined in (11).
6 Compute $\hat{t} = \hat{Q}_{1-\alpha}(\{E_i\}_{i \in \mathcal{D}^{\text{cal}}})$ as the $\lceil (1 - \alpha)(1 + |\mathcal{D}^{\text{cal}}|) \rceil$th smallest value in $\{E_i\}_{i \in \mathcal{D}^{\text{cal}}}$.
7 Select the $\hat{t}$-th element from $\{S_t(X_{n+1}, \varepsilon_{n+1}, \hat{\pi})\}_{t=0}^T$, defined in (8)–(10):

$$\hat{C}_{n,\alpha}^{\text{sc}}(X_{n+1}) = S_{\hat{t}}(X_{n+1}, \varepsilon_{n+1}, \hat{\pi}).$$

8 **Output:** A prediction interval $\hat{C}_{n,\alpha}^{\text{sc}}(X_{n+1})$ for $Y_{n+1}$.

---

**Theorem 1** (Marginal coverage)**.** *If $(X_i, Y_i)$, for $i \in \{1, \ldots, n + 1\}$, are exchangeable, then the output of Algorithm 1 satisfies:*

$$\mathbb{P}\left[Y_{n+1} \in \hat{C}_{n,\alpha}^{\text{sc}}(X_{n+1})\right] \geq 1 - \alpha. \tag{12}$$

Note that Theorem 1 provides only a lower bound; a nearly matching upper bound on the marginal coverage can be generally established for split-conformal inference if the conformity scores are almost-surely distinct [24, 30, 35]. Although the CHR scores (11) are discrete, our experiments will show the coverage is tight as long as the resolution $T$ is not too small.

## 3  Asymptotic analysis

We prove here that the prediction intervals computed by CHR (Algorithm 1) are asymptotically equivalent, as $n \to \infty$, to those of the oracle from (3)–(4), if the model $\hat{\pi}$ is consistent for $P_{Y|X}$ and a few other technical conditions are met. In particular, we analyze a slightly modified version of Algorithm 1 in which there is no randomization; this is theoretically more amenable and equivalent in spirit, although it may yield wider intervals in finite samples. Our theory relies on the additional Assumptions 1–5, explained below and stated formally in Supplementary Section S2.

1. The samples are i.i.d., which is stronger than exchangeability; this is the key to our concentration results.

2. The black-box model estimates $P_{Y|X}$ consistently, in a sense analogous to that in [24, 32]. This assumption is crucial and may be practically difficult to validate in practice, but it

can be justified by existing consistency results available for some models under suitable conditions, such as random forests [27]. Further, the resolution $m$ of the partition of the $Y$ domain should grow with $n$ at a certain rate, and the resolution $T$ of the scores $E_i$ in (11) should grow as $T_n = n$.

3. The true $P_{Y|X}$ is continuous and with bounded density within a finite domain. This assumption is technical and could be relaxed with more work.

4. The true $P_{Y|X}$ is unimodal. This assumption is also technical and could be relaxed with more work.

5. The estimated conditional histogram $\hat{\pi}$ preserves the boundedness and unimodality of $P_{Y|X}$; this assumption may be unnecessary but it is convenient and quite innocuous at this point given Assumptions 2–4.

For simplicity, we assume the number of observations is $2n$, the test point is $(X_{2n+1}, Y_{2n+1})$, and $\mathcal{D}^{\mathrm{train}} = \mathcal{D}^{\mathrm{cal}} = n$, although different relative sample sizes would yield the same results.

**Theorem 2** (Asymptotic conditional coverage and optimality). *$\forall \alpha \in (0, 1]$, let $\hat{C}^{\mathrm{sc}}_{n,\alpha}(X_{2n+1})$ denote the prediction interval for $Y_{2n+1}$ computed by Algorithm 1 at level $1 - \alpha$ without randomization. Under Assumptions 1–5, $\hat{C}^{\mathrm{sc}}_{n,\alpha}(X_{2n+1})$ is asymptotically equivalent, as $n \to \infty$, to $C^{\mathrm{oracle}}_{\alpha}(X_{2n+1})$, the output of the oracle* (3)–(4). *In particular, the following two properties hold.*

*(i) Asymptotic oracle length. For some sequences $\gamma_n \to 0$ and $\xi_n \to 0$ as $n \to \infty$,*

$$\mathbb{P}\Big[|\hat{C}^{\mathrm{sc}}_{n,\alpha}(X_{2n+1})| \leq |C^{\mathrm{oracle}}_{\alpha}(X_{2n+1})| + \gamma_n\Big] \geq 1 - \xi_n.$$

*(ii) Asymptotic conditional coverage. For some sequences $\epsilon_n \to 0$ and $\zeta_n \to 0$ as $n \to \infty$,*

$$\mathbb{P}\Big[\mathbb{P}\Big[Y \in \hat{C}^{\mathrm{sc}}_{n,\alpha}(X_{2n+1}) \mid X_{2n+1}\Big] \geq 1 - \alpha - \epsilon_n\Big] \geq 1 - \zeta_n.$$

Theorem 2 is similar to results in [24] and [32] about the efficiency of earlier approaches to conformal regression, including CQR [32]. However, the increased flexibility of our method is reflected by the oracle in Theorem 2, which is stronger than those in [24, 32]. In fact, the oracle in [24] does not have conditional coverage, and that in [32] produces wider prediction intervals with constant lower and upper miscoverage rates. Other conformal methods based on non-parametric density estimation [10, 21] are not as efficient as CHR, in the sense that Theorem 2 does not hold for them.

## 4  Numerical experiments

### 4.1  Software implementation

A Python implementation of CHR is available online at `https://github.com/msesia/chr`, along with code to reproduce the following numerical experiments. This software divides the domain of $Y$ into a desired number of bins with equal sizes, depending on the range of values observed in the training data; we use 100 bins for the synthetic data and 1000 for the real data. Then, we estimate the conditional histograms $\hat{\pi}$ using different black-box quantile regression models [27, 33], with a grid of quantiles ranging from 1% to 99%; see Supplementary Section S1.1. Our software also supports Bayesian additive regression trees [11] and could easily accommodate other alternatives. For simplicity, we apply CHR and other benchmark methods by assigning equal numbers of samples to the training and calibration sets; this ensures all comparisons are fair, although different options may lead to even shorter intervals [32]. See [8] for a rigorous discussion of how this choice affects the variability of the coverage conditional on the calibration data, which is an issue we do not explore here. See Supplementary Section S3 for details about how the models are trained, and information about the necessary computational resources.

### 4.2  Synthetic data

We simulate a synthetic data set with a one-dimensional feature $X$ and a continuous response $Y$, from the same distribution previewed in Figure 1, which is similar to that utilized in [30] to present CQR. Our method is applied to 2000 independent observations from this distribution, using the first 1000 of

them for training a deep quantile regression model, and the remaining ones for calibration. Figure 1 visualizes the resulting prediction bands for independent test data, comparing them to the analogous quantities output by CQR. Both methods are based on the same neural network and guarantee 90% marginal coverage, but ours leads to narrower intervals. Indeed, the advantage of CHR is that it can extract information from all conditional quantiles estimated by the base model and then automatically adapt to the estimated data distribution. By contrast, CQR [30] can only leverage a pre-specified lower and upper quantile (e.g., 5% and 95% in this example), and is therefore not adaptive to skewness.

Figure 3 (a) summarizes the performance of CHR over 100 experiments based on independent data sets, as a function of the sample size. We evaluate the marginal coverage, approximate the worst-slab conditional coverage [9] as in [31], and compute the average interval width. We consider two benchmarks in addition to CQR [30]: distributional conformal prediction (DCP) [10] and DistSplit [21]. To facilitate the comparisons, all methods have the same base model. (We also applied DistSplit as implemented by [21], with a different base model, but the version presented here performs better.) These results show CHR leads to the shortest prediction intervals, while simultaneously achieving the highest conditional coverage. Compatibly with Theorem 2, the output of CHR becomes roughly equivalent to that of the omniscient oracle as the sample size grows; the latter can be implemented exactly here because we know the true data generating process.

Supplementary Figure S1 compares the performance of CHR in these experiments to that of naive uncalibrated 90% prediction intervals based on the same deep neural network regression model, and obtained by simply plugging $\hat{\pi}$ into the oracle in (7), with $\tau = 0.9$. Unsurprisingly, the naive prediction intervals do not generally have the desired marginal coverage; in this case, they tend to be too narrow if the sample size is small and too wide if the sample size is large. Although the lack of coverage for the uncalibrated intervals is not very pronounced here because this black-box model is relatively accurate even with $n = 100$, such naive approach can yield arbitrarily low coverage in general, especially if the learning task is more difficult (e.g., for high-dimensional $X$), and is thus not reliable.

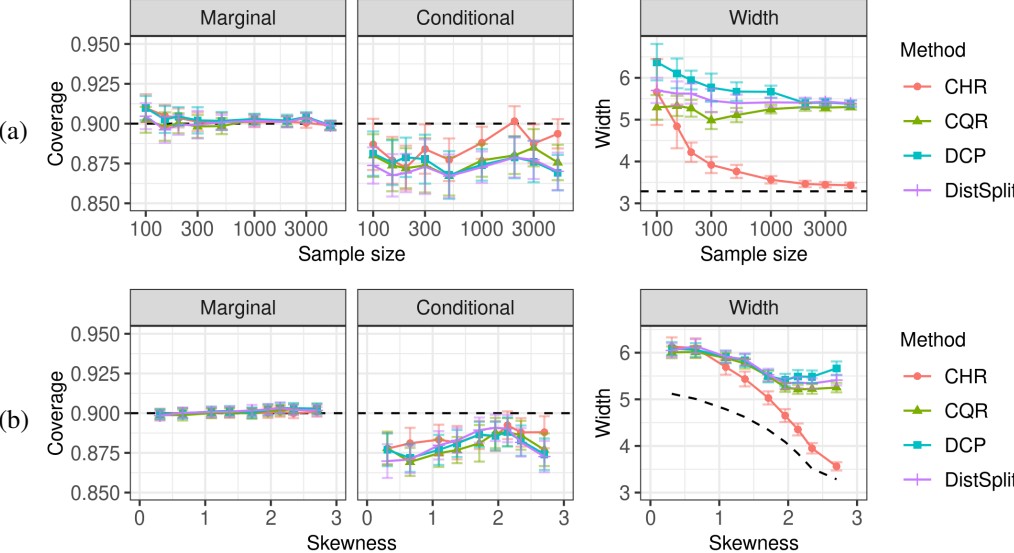

Figure 3: Performance of our method (CHR) and benchmarks on synthetic data distributed as in Figure 1. The dashed lines and curves correspond to an omniscient oracle. The vertical error bars span two standard errors from the mean. (a) Performance vs. sample size. (b) Performance vs. average skewness of the conditional distribution of the response, with a sample size of 1000. The maximum skewness (near 3) matches that of the data in (a).

Figure 3 (b) shows analogous results from experiments in which we fix the sample size to 1000 and vary instead the skewness of the data distribution. Precisely, we flip a biased coin for each data point and transform $Y$ into $-Y$ if it lands heads, varying the coin bias as a control parameter. At one end of this spectrum, we recover the same skewed data distribution as in Figure 3 (a); at the other end, we have a symmetric $P_{Y|X}$. Our results are reported as a function of the expected skewness, defined

as $\mathbb{E}[(Y - \mu(X))^3/\sigma^3(X)]$, where $\mu(X)$ and $\sigma(X)$ are the mean and standard deviation of $Y \mid X$, respectively. These experiments show all methods are equivalent in terms of interval length if $P_{Y|X}$ is symmetric (skewness close to 0), while CHR can be much more powerful if $P_{Y|X}$ is skewed.

## 4.3 Real data

We apply CHR to the following seven public-domain data sets also considered in [30]: physicochemical properties of protein tertiary structure (bio) [6], blog feedback (blog) [1], Facebook comment volume [2], variants one (fb1) and two (fb2), from the UCI Machine Learning Repository [15]; and medical expenditure panel survey number 19 (meps19) [3], number 20 (meps20) [4], and number 21 (meps21) [5], from [13]. We refer to [30] for more details about these data. As in the previous section, we would like to compare CHR to CQR, DistSplit, and DCP. However, as DCP [10] is unstable on all but one of these data sets, sometimes leading to very wide intervals, we replace it instead with a new hybrid benchmark that we call DCP-CQR. This improves the stability of DCP by combining it with CQR [30], as explained in Supplementary Section S1.3. This limitation of DCP may be explained by noting the method needs to learn a reasonably accurate approximation of the full conditional distribution of $Y \mid X$, and its performance is particularly sensitive to the estimation of the tails, which is most difficult; see Supplementary Section S1.3 for more details. By contrast, CHR is robust because it only needs to estimate a histogram with relatively few bins—a much easier statistical task—and then it specifically focuses on finding the shortest intervals containing high probability mass. We apply all methods, including our CHR, based on the same deep quantile regression model. Their performances are evaluated as in the previous section, averaging over 100 independent experiments per data set. In each experiment, 2000 samples are used for training, 2000 for calibration, and the remaining ones for training. All features are standardized to have zero mean and unit variance. The nominal coverage rate is 90%.

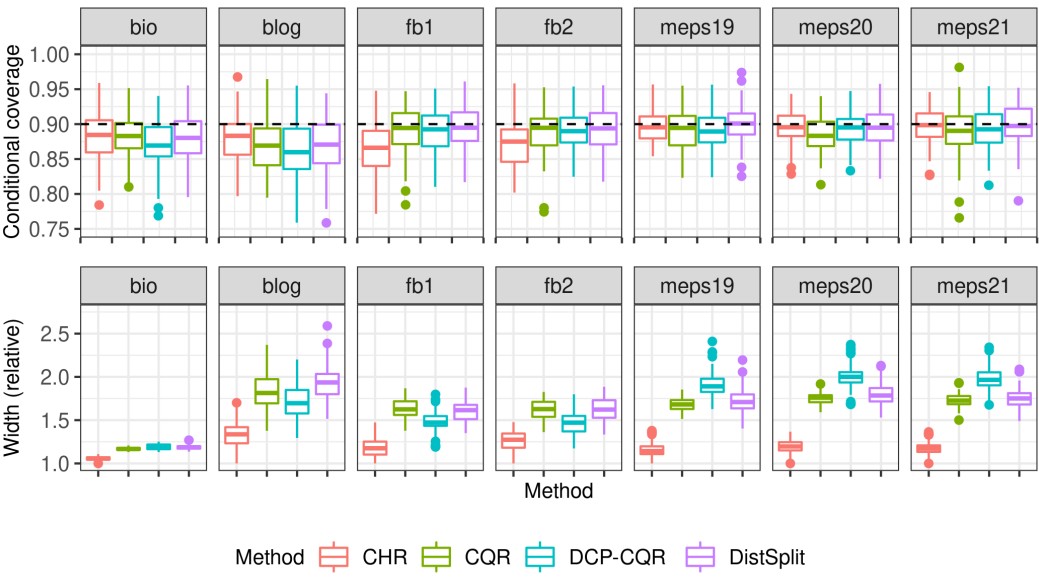

Figure 4: Performance of our method (CHR) and benchmarks on several real data sets, using a deep neural network model. All methods provably have 90% marginal coverage. The box plots show the distribution of results over 100 random test sets, each containing 2000 observations.

Figure 4 shows the distribution of the conditional coverage and interval width corresponding to different methods, separately for each data set. To simplify the plots by using a shared vertical axis, the widths of the prediction intervals are scaled, separately for each data set, so that the smallest one is always equal to one. Marginal coverage is omitted here because all methods provably control it; however, it can be found in Supplementary Table S1. All methods perform well in terms of worst-slab conditional coverage. CHR outperforms the others in terms of statistical efficiency, as it consistently leads to the shortest intervals. CQR and DistSplit are comparable to each other, while DCP-CQR sometimes outputs wider intervals. Supplementary Figure S2 shows that analogous results

are obtained if we utilized a random forest model instead of a deep neural network. Supplementary Table S1 summarizes these results in more detail, including marginal coverage and the omitted performance of the original DCP. Finally, Supplementary Figure S3 compares the performance of CHR in these experiments with real data to that of naive uncalibrated 90% prediction intervals based on the same deep neural network regression model, as in Figure S1. These results show that the naive prediction intervals do not generally have the desired marginal coverage; in some cases they are too narrow, and in others they are too wide.

## 5    Conclusions

This paper developed CHR, a non-parametric regression method based on novel conformity scores leading to shorter prediction intervals with coverage, and enjoying stronger asymptotic efficiency, compared to the state-of-the-art alternatives. Of course, real data are finite and our theory relies on assumptions which may be difficult to validate; nonetheless, it is a sanity check and it provides an informative comparison. Further, the experiments confirm CHR performs well in practice.

The ability of CHR to automatically adapt to unknown skewness may prove useful in practice, as empirical data often follow distributions with power-law tails [12]. Indeed, the data sets analyzed in Section 4.3 tend to have highly skewed outcomes with many observations equal to zero. At the same time, a limitation of CHR is that it does not rigorously control the lower and upper miscoverage rates, which may be important for some applications; if that is the case, the modified version of CQR proposed by [30] would be a better choice. Note that the standard implementations of CQR and of the other benchmarks [10, 21] considered in this paper are not guaranteed to separately control the lower and upper miscoverage rates. In any case, users of our method could naturally obtain approximations of the lower and upper miscoverage rates for any prediction interval by looking at the underlying conditional histograms, although these estimates are of course not calibrated in finite samples.

Algorithm S1 in Supplementary Section S1.4 extends CHR to accommodate cross-validation+ [7], which is often more powerful, and computationally expensive, compared to the split-conformal approach presented in this paper. The strategy is the same as that followed by [31] in the classification setting, although it requires an extra step, in which the standard cross-validation+ prediction set [7] is replaced by its convex hull to guarantee the final output is an interval [17]. Supplementary Theorem S1 establishes that Algorithm S1 leads to marginal coverage above $1 - 2\alpha$, applying the more general theory from [17]. Finally, a possible directions for future research may involve the extension of our method to deal with multi-dimensional responses $Y$.

From a broader perspective, this paper falls within a rapidly growing body of works focusing on improving the interpretability and statistical reliability of machine learning algorithms. Prediction intervals with marginal coverage provide a convenient way of communicating uncertainty about the accuracy of any machine learning black-box, which is important to increase their reliability, to ensure their fairness [29], and to facilitate their acceptance. Furthermore, conformal prediction intervals provide a principled metric by which to compare different machine learning algorithms [18].

### Acknowledgments and Disclosure of Funding

The authors are grateful to Stephen Bates, Emmanuel Candès, and Wenguang Sun for providing insightful comments about an earlier version of this manuscript. M.S. thanks the center for Advanced Research Computing at the University of Southern California for providing computing resources. Y.R. was supported by the Israel Science Foundation (grant 729/21). Y.R. also thanks the Career Advancement Fellowship, Technion, for providing research support.

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
