# Supplementary Material for
# Conformal Prediction using Conditional Histograms

**Matteo Sesia**
Department of Data Sciences and Operations
University of Southern California, USA
sesia@marshall.usc.edu

**Yaniv Romano**
Department of Electrical and Computer Engineering and
Department of Computer Science
Technion, Israel
yromano@technion.ac.il

## S1 Supplementary methods

### S1.1 Estimating conditional distributions and histograms

For any fixed $K > 1$, define the sequence $a_k = k/K$ for $k \in \{0, \ldots, K\}$, and let $\hat{q}(x) = (\hat{q}_{a_0}(x), \ldots, \hat{q}_{a_K}(x))$ denote a collection of $K+1$ conditional quantile estimators,[1] where $\hat{q}_{a_k}(x)$ attempts to approximate the $a_k$-th quantile of the conditional distribution of $Y \mid X = x$, such that $\hat{q}_{a_k}(x) \leq \hat{q}_{a_{k+1}}(x)$ for all $k$ and $x$. Note that we allow multiple estimated quantiles to be identical to each other, to accommodate the possibility of point masses. Furthermore, we assume $\hat{q}_0(x)$ and $\hat{q}_1(x)$ are conservative upper and lower bounds for the support of $Y \mid X = x$, i.e., $\hat{q}_0(X) = b_0 < Y < b_m = \hat{q}_1(X)$. We will discuss in the next section practical options for estimating $\hat{q}(x)$. Now, we leverage any given $\hat{q}(x)$ to compute estimates $\hat{\pi}_j(x)$ of the unknown bin probabilities $\pi_j(x)$ in (6), for all $j \in \{1, \ldots, m\}$. Although there are multiple way of doing this, a principled solution is to convert the information contained in $\hat{q}$ into a piece-wise constant density estimate, and then integrate that density within each bin. Precisely, for any fixed $x$, let $\hat{c}(x) = (\hat{c}_0(x), \ldots, \hat{c}_{\bar{m}(x)}(x))$ denote the strictly increasing sequence of $\bar{m}(x) \leq m$ unique values in $\hat{q}(x)$, and define our estimated conditional density $\hat{f}$ as

$$\hat{f}(y \mid x) = \frac{1}{\bar{m}(x)} \sum_{j=1}^{\bar{m}(x)} h_j(x) \mathbb{1}\left[\hat{c}_{j-1}(x) < y < \hat{c}_j(x)\right],$$

with

$$h_j(x) = \frac{\#\{j' \in \{0, \ldots, m\} : \hat{q}_{a_{j'}}(x) = \hat{c}_j(x)\}}{m \cdot [\hat{c}_j(x) - \hat{c}_{j-1}(x)]}.$$

Intuitively, $\hat{f}$ is a histogram with $\bar{m}(x)$ bins, whose delimiters are $(\hat{c}_0(x), \ldots, \hat{c}_{\bar{m}(x)}(x))$ and whose heights are $(h_1(x), \ldots, h_{\bar{m}(x)}(x))$. The numerator in the expression for $h_j(x)$ counts the number of estimated quantiles that are identical to the $\hat{c}_j$-th one, accounting for the possible presence of point masses in the approximation of $P_{Y|X}$ captured by $\hat{q}(x)$.

As the tails of the above estimated conditional density may be particularly inaccurate because relatively little information is available to estimate extremely low or high quantiles, we smooth them.

---

[1] Recall the definition of conditional quantiles: each $\hat{q}_c(x)$ is an estimate of the true $c$-th conditional quantile of $Y \mid X = x$: that is, the smallest value of $y$ such that $\mathbb{P}[Y \leq y \mid X = x] \geq c$.

35th Conference on Neural Information Processing Systems (NeurIPS 2021).

This ensures any estimation errors will not make $\hat{f}$ decay too fast, forcing one to look much farther than necessary in the tails before finding sufficient mass for the desired prediction intervals. The smoothing approach we adopt simply consists of making $\hat{f}$ constant between $b_0$ (the uniform lower bound on $Y$) and the 1% quantile, as well as between the 99% quantile and $b_m$ (the uniform upper bound on $Y$), distributing these 1% probability masses uniformly in the tails.[2]

We utilize the same estimated conditional distribution thus obtained for our method as well as for our implementations of DCP and DistSplit, because it performs relatively well for all of them. In particular, our method leverages this distribution to construct a conditional histogram as follows. The probability mass $\hat{\pi}_j(x)$ for the bin $[b_{j-1}, b_j)$ is given by:

$$\hat{\pi}_j(x) = \int_{b_{j-1}}^{b_j} \hat{f}(y \mid x)dy, \tag{S1}$$

which is easy to compute because $\hat{f}$ is piece-wise constant. Finally, $\sum_{j=1}^{m} \hat{\pi}_j(x) = 1$ by construction.

When implemented with a deep neural network [8], the multi-quantile regression method described above has computational cost comparable to that of the bi-quantile regression model utilized by CQR [9]. Indeed, the numbers of parameters and the architecture of the neural network are essentially the same in both cases, the only difference is that our model has a wider output layer. Therefore, the computational cost and training runtime are approximately the same. Intuitively, this can be understood as thinking of the neural network as learning an approximate representation of the conditional distribution of $Y \mid X$, regardless of how many different quantiles are explicitly estimated. Of course, that is not to say that estimating many quantiles is as easy as estimating only two, but most of the additional statistical difficulty would come from estimating extremely large or small quantiles, not the intermediate ones. Precisely to avoid this problem, our model does not attempt to estimate extremely large or small quantiles (below 1% or above 99%); instead, the tails are smoothed as explained above.

## S1.2    Randomized prediction intervals

Due to the discrete nature of the optimization problem in (7), the inequality involving $\tau$ may not be binding at the optimal solution. Therefore, to avoid producing wider intervals than necessary, we introduce some randomization. Let $\varepsilon$ be a uniform random variable between 0 and 1 drawn independently of everything else. Then, define the following function $R$, which takes as input $[l, u] \subseteq \{1, \dots, m\}$, $x, \pi, \varepsilon, \tau$, and outputs a sub-interval of $\{1, \dots, m\}$:

$$R([l, u], x, \varepsilon, \pi, \tau) := \begin{cases} [l, u], & \text{if } \varepsilon > V([l, u], x, \pi, \tau), \\ [l-1, u], & \text{if } \varepsilon \le V([l, u], x, \pi, \tau) \text{ and } \pi_l(x) \le \pi_u(x), \\ [l, u-1], & \text{if } \varepsilon \le V([l, u], x, \pi, \tau) \text{ and } \pi_l(x) > \pi_u(x), \end{cases} \tag{S2}$$

where the function $V$ is given by

$$V([l, u], x, \pi, \tau) := \frac{\sum_{j=l}^{u} \pi_j(x) - \tau}{\min\{\pi_l(x), \pi_u(x)\}}.$$

In words, $R$ returns a random subset of $[l, u]$ by removing the extreme bin with the smallest mass according to $\pi$, based on the outcome of a biased coin flip, if the total mass in the original interval exceeds $\tau$. Consequently, the total mass in $R([l, u], x, \varepsilon, \pi, \tau)$ will on average be exactly equal to $\tau$ if $[l, u]$ is given by (7).

Inspired by the above oracle, the randomized version of our algorithm is implemented as follows. First, fix any *starting point* $\bar{t} \in \{0, \dots, T\}$ and define $S_{\bar{t}}$ by applying (7) and (S2) without the nesting constraints (with $S^- = \emptyset$ and $S^+ = \{1, \dots, m\}$):

$$S_{\bar{t}}^0 := \mathcal{S}(x, \pi, \emptyset, \{1, \dots, m\}, \tau_{\bar{t}}), \qquad\qquad S_{\bar{t}} := R(S_{\bar{t}}^0, x, \varepsilon, \pi, \tau_{\bar{t}}). \tag{S3}$$

Having computed the initial interval $S_t$ for $t = \bar{t}$, we recursively extend the definition to the wider intervals indexed by $t = \bar{t} + 1, \dots, T$ as follows:

$$S_t^0 := \mathcal{S}(x, \pi, S_{t-1}, \{1, \dots, m\}, \tau_t),$$
$$S_t := \begin{cases} R(S_t^0, x, \varepsilon, \pi, \tau_t), & \text{if } S_{t-1} \subseteq R(S_t^0, x, \varepsilon, \pi, \tau_t), \\ S_t^0, & \text{otherwise.} \end{cases} \tag{S4}$$

---

[2]We thank Stephen Bates for suggesting a smoothing strategy which inspired this solution.

Intuitively, the randomization step in (S4) is applied only if it does not violate the nesting constraints, ensuring $S_{\bar{t}} \subseteq S_{\bar{t}+1} \subseteq \ldots \subseteq S_T$. See the top row of Figure 2 for a schematic of this step. Similarly, the narrower intervals $S_t$ indexed by $t = \bar{t} - 1, \bar{t} - 2, \ldots 0$ are defined recursively as:

$$S_t^0 := \mathcal{S}(x, \pi, \emptyset, S_{t+1}^0, \tau_t), \qquad\qquad S_t := R(S_t^0, x, \varepsilon, \pi, \tau_t). \qquad\qquad \text{(S5)}$$

See the bottom row of Figure 2 for a schematic of this step. Note that $\mathcal{S}$ in (7) is applied here in (S5) with $S^+ = S_{t+1}^0$ to ensure the optimization problem has a feasible solution; this may not necessarily be the case with $S^+ = S_{t+1}$, as the latter is randomized and may therefore sometimes contain less mass than necessary, according to the input $\pi$. Nonetheless, the sequence of intervals $\{S_t\}_{t=0}^T$ thus obtained is provably nested, as previewed in Figure 2. In the following, it will be convenient to highlight the dependence of this sequence on $x, \varepsilon, \pi$ by writing it as $S_t(x, \varepsilon, \pi)$.

**Proposition 1.** *The sequence of intervals $\{S_t\}_{t=0}^T$ defined recursively by (S3)–(S5), and depending on $x, \varepsilon, \pi$, always satisfies $S_{t-1} \subseteq S_t$ for all $t \in \{1, \ldots, T\}$.*

Proposition 1 is proved below. Again, note that this results holds regardless of the starting point $\bar{t}$ in (S3), although the most intuitive choice is to pick $\bar{t}$ such that $\tau_{\bar{t}} \approx 1 - \alpha$.

*Proof of Proposition 1.* First, we show $S_{t-1} \subseteq S_t$ for all $t = \bar{t} + 1, \ldots, T$. We know from (S4) that there are two possibilities. (i) If $S_{t-1} \subseteq R(S_t^0, x, \varepsilon, \pi, \tau_t)$, then $S_t = R(S_t^0, x, \varepsilon, \pi, \tau_t)$ and so $S_{t-1} \subseteq S_t$. (ii) Otherwise, $S_t = S_t^0 = \mathcal{S}(x, \pi, S_{t-1}, \{1, \ldots, m\}, \tau_t)$, which contains $S_{t-1}$ by definition of $\mathcal{S}$ in (7).

Second, we show $S_t \subseteq S_{t+1}$ for all $t = \bar{t} - 1, \ldots, 0$, using (S5). Here, we can also distinguish between two possibilities. (i) If $S_{t+1} = S_{t+1}^0$, then $S_t^0 \subseteq S_{t+1}$ by definition of $\mathcal{S}$ in (7), and so $S_t \subseteq S_{t+1}$ because $S_t \subseteq S_t^0$. (ii) Otherwise, $S_{t+1}$ must have been randomized and this is the least obvious case on which we focus below.

Suppose $S_{t+1} \subset S_{t+1}^0$. We know from the definition of $\mathcal{S}$ in (7) that $S_t^0 \subseteq S_{t+1}^0$. On the one hand, if $S_t^0 = S_{t+1}^0$, it is easy to see from (S2) that $S_t = S_{t+1}$ because $\tau_t < \tau_{t+1}$, and so the same bin randomly removed from $S_{t+1}^0$ will also certainly be removed from $S_t$. On the other hand, if $S_t^0 \subset S_{t+1}^0$, it must be the case that $S_t^0 \subseteq S_{t+1}$ because $S_{t+1}$ is obtained by removing the boundary bin of $S_{t+1}^0$ with the smallest mass. Therefore, $S_t^0$ cannot include the aforementioned bin without also satisfying $S_t^0 = S_{t+1}^0$, for otherwise it would be possible to find an alternative $S_t^{0'}$ with equal length and smaller but still feasible mass above $\tau_t$, which is inconsistent with optimality of $S_t^0$ according to the definition of $\mathcal{S}$ in (7). This implies $S_t \subset S_{t+1}^0$ because $S_t \subset S_t^0$, completing the proof. $\qquad\square$

### S1.3 The DCP-CQR hybrid method

The DCP-CQR hybrid repurposes the DCP calibration algorithm [3] to adaptively choose which lower and upper estimated quantiles should be extracted from the machine-learning model; then, it takes these as a starting point for CQR [9]. By contrast, the original CQR requires one to pre-specify which two conditional quantiles should be estimated by the machine learning model. For example, we implement CQR by estimating the $\alpha/2$ and $1 - \alpha/2$ quantiles, as this is the most intuitive choice and it guarantees the method is asymptotically efficient [11], although in a weaker sense compared to the oracle property established by Theorem 2 for CHR.

The reason why DCP-CQR is more stable than DCP is that our hybrid only considers a limited grid of possible quantiles (e.g., 1% to 99%). If the machine learning model is very inaccurate and the fixed quantile grid turns out to be insufficient to reach 90% coverage (assuming $\alpha = 0.1$) on the calibration data, then we can simply rely on CQR to correct the coverage by adding a constant shift to the prediction bands. By contrast, the original DCP [3] may sometimes rely on extreme quantiles (e.g., 99.99%) of the conditional distribution estimated by the fitted model, which are unreliable.

### S1.4 Calibration with cross-validation+

Algorithm S1 extends Algorithm 1 to accommodate a calibration scheme alternative to data splitting: cross-validation+ [1]. While we do not fully review cross-validation+ for lack of space, readers aware of the work of [1], or [4], will recognize this as a straightforward combination of their techniques with our novel conformity scores.

---

**Algorithm S1:** CV+ adaptive predictive intervals for regression

---

**1 Input:** data $\{(X_i, Y_i)\}_{i=1}^n$, $X_{n+1}$, partition $\mathcal{B}$ of the domain of $Y$, level $\alpha \in (0,1)$, resolution $T$ for the conformity scores, starting index $\bar{t}$ for recursive definition of conformity scores, machine-learning algorithm for estimating conditional distributions.

**2** Randomly split the training data into $K$ disjoint subsets, $\mathcal{D}_1, \ldots, \mathcal{D}_K$, each of size $n/K$.

**3** Sample $\varepsilon_i \sim \text{Uniform}(0,1)$ for each $i \in \{1, \ldots, n+1\}$, independently of everything else.

**4 for** $k \in \{1, \ldots, K\}$ **do**

**5** $\quad$ Train any estimate $\hat{\pi}^k$ of the mass of $Y \mid X$ for each bin in $\mathcal{B}$, e.g., with (S1), based on all data points except those in $\mathcal{D}_k$.

**6 end**

**7** Use the function $E$ defined in (11) to construct the prediction interval

$$\hat{C}_{n,\alpha}^{\text{CV+}}(X_{n+1}) = \text{Conv}\,(C), \tag{S6}$$

where $\text{Conv}(C)$ is the convex hull of the set $C$, which is defined as

$$C = \left\{ y : \frac{1}{n} \sum_{i=1}^n \mathbf{1}\left[ E(X_i, Y_i, \varepsilon_i, \hat{\pi}^{k(i)}) < E(X_{n+1}, y, \varepsilon_{n+1}, \hat{\pi}^{k(i)}) \right] < 1 - \alpha_n \right\}, \tag{S7}$$

with $\alpha_n = \alpha(1 + 1/n) - 1/n$ and $k(i) \in \{1, \ldots, K\}$ is the fold containing the $i$-th sample.

**8 Output:** A prediction interval $\hat{C}_{n,\alpha}^{\text{CV+}}(X_{n+1})$ for the unobserved label $Y_{n+1}$.

---

**Theorem S1** (Adapted from Theorem 3 in [4]). *Under the same assumptions of Theorem 1, if $\hat{\pi}$ is invariant to permutations of its input samples, the output of Algorithm S1 satisfies:*

$$\mathbb{P}\left[ \{Y_{n+1} \in \hat{C}_{n,\alpha}^{\text{CV+}}(X_{n+1})\} \right] \geq 1 - 2\alpha - \min\left\{ \frac{2(1 - 1/K)}{n/K + 1}, \frac{1 - K/n}{K + 1} \right\}. \tag{S8}$$

*In the special case where $K = n$, this bound simplifies to:*

$$\mathbb{P}\left[ Y_{n+1} \in \hat{C}_{n,\alpha}^{\text{JK+}}(X_{n+1}) \right] \geq 1 - 2\alpha. \tag{S9}$$

## S2 Theoretical analysis

### S2.1 Finite-sample analysis

*Proof of Theorem 1.* The interval $\hat{C}_{n,\alpha}^{\text{sc}}(X_{n+1})$ is such that $Y_{n+1} \in \hat{C}_{n,\alpha}^{\text{sc}}(X_{n+1})$ if and only if

$$\min\{t \in \{0, \ldots, T\} : Y_{n+1} \in S_t(X_{n+1}, \varepsilon_{n+1}, \hat{\pi})\} \leq \hat{Q}_{1-\alpha}(\{E_i\}_{i \in \mathcal{D}^{\text{cal}}}).$$

Equivalently, $Y_{n+1} \in \hat{C}_{n,\alpha}^{\text{sc}}(X_{n+1})$ if and only if

$$E_{n+1} \leq \hat{Q}_{1-\alpha}(\{E_i\}_{i \in \mathcal{D}^{\text{cal}}}). \tag{S10}$$

The proof is standard from here: the key idea is that the probability of the event in (S10) is at least $1 - \alpha$ because all conformity scores $\{E_i\}_{i=1}^{n+1}$ are exchangeable; see [9] for details. $\qquad\square$

### S2.2 Asymptotic analysis

**Assumption 1** (i.i.d. data). *The data $\{(X_i, Y_i)\}_{i=1}^{2n+1}$ are i.i.d. from some unknown joint distribution.*

**Assumption 2** (consistency). *For any fixed $n$, let $m_n$ denote the number of bins in the partition $\mathcal{B}$ of the space of $Y$ utilized by our method. Let $F(y \mid x)$ denote the cumulative distribution function of $Y \mid X = x$, and define $\hat{F}(y \mid x)$ as the estimate of the latter according to $\hat{\pi}$ from (S1); i.e.,*

$$\hat{F}(y \mid x) := \sum_{j=1}^{\hat{j}(y)} \hat{\pi}_j(x),$$

*where $\hat{j}(y) = \max\{j \in \{1, \ldots, m_n\} : y \leq b_j\}$. Then, assume there exists a sequence $\eta_n \to 0$, as $n \to \infty$, such that, for all $j \in \{1, \ldots, m_n\}$,*

$$\mathbb{P}\left[\mathbb{E}\left[\left(\hat{F}(b_j \mid X) - F(b_j \mid X)\right)^2 \mid \mathcal{D}^{\mathrm{train}}\right] \leq \eta_n^2\right] \geq 1 - \eta_n^2. \tag{S11}$$

*Further, $m_n = \lfloor \eta_n^{-1} \rfloor$ and $T_n = n$, where $T_n$ is the resolution of the conformity scores $E_i$ (11).*

**Assumption 3** (regularity). *For any $x \in \mathbb{R}^p$, the conditional distribution of $Y \mid X = x$ is continuous with density $f(y \mid x)$ and support $[-C, C]$, for some finite $C > 0$. Furthermore, $1/K < f(y \mid x) < K/2$ within $[-C, C]$, for some $K > 0$.*

**Assumption 4** (unimodality). *For any $x \in \mathbb{R}^p$, the conditional distribution of $Y \mid X = x$ is unimodal; i.e., there exists $y_0 \in [-C, C]$ (depending on $x$), such that $f(y_0 + y'' \mid x) \leq f(y_0 + y')$ if $y'' \geq y' \geq 0$, and $f(y_0 + y'' \mid x) \leq f(y_0 + y')$ if $y'' \leq y' \leq 0$.*

**Assumption 5** (smoothing). *For any fixed $n$ and $x \in \mathbb{R}^p$, the estimated conditional distribution of $Y \mid X = x$ characterized by $\hat{\pi}(x)$ is unimodal. That it, there exists $j_0 \in \{1, \ldots, m_n\}$ such that $\hat{\pi}_{j_0+k''}(x) \leq \hat{\pi}_{j_0+k'}(x)$ if $k'' \geq k' \geq 0$, and $\hat{\pi}_{j_0+k''}(x) \leq \hat{\pi}_{j_0+k'}(x)$ if $y'' \leq y' \leq 0$, for all $k'', k'$ such that $j_0 + k'' \in \{1, \ldots, m\}$ and $j_0 + k' \in \{1, \ldots, m\}$. Furthermore, assume $\hat{\pi}_j \leq K$ for all $j \in \{1, \ldots, m_n\}$, for any $n$.*

Note that, if $\hat{\pi}$ is based on a quantile model as described in Section S1.1, Assumption 2 is closely related to the consistency assumption on the estimated conditional quantiles utilized by [11] to study CQR [9], although the latter only involved two fixed quantiles. More precisely, leveraging Assumption 3, one could rewrite (S11) in terms of the consistency of the underlying quantile regressors,

$$\mathbb{P}\left[\mathbb{E}\left[(\hat{q}_{\tau_t}(X) - q_{\tau_t}(X))^2 \mid \mathcal{D}^{\mathrm{train}}\right] \leq \tilde{\eta}_n\right] \geq 1 - \tilde{\rho}_n, \tag{S12}$$

for some sequences $\tilde{\eta}_n \to 0$ and $\tilde{\rho}_n \to 0$ as $n \to \infty$. The assumption in (S12) is also similar to that adopted in [6] for mean regression estimators, and it is weaker than requiring consistency in the sense of $L^2$ convergence, by Markov's inequality.

**Main result**

**Theorem S2** (More precise restatement of Theorem 2). *For any $\alpha \in (0, 1]$, let $\hat{C}_{n,\alpha}^{\mathrm{sc}}(X_{2n+1})$ denote the prediction interval at level $1 - \alpha$ for $Y_{2n+1}$ obtained by applying Algorithm 1 with $\varepsilon_i = 0$ for all $i \in \{n+1, \ldots, 2n+1\}$; that is, we omit the randomization in (S2). Under Assumptions 1–5, the prediction interval $\hat{C}_{n,\alpha}^{\mathrm{sc}}(X_{2n+1})$ is asymptotically equivalent, as $n \to \infty$, to $C_\alpha^{\mathrm{oracle}}(X_{2n+1})$—the output of the ideal oracle from (3)–(4). In particular, the following two properties hold.*

*(i) Asymptotic oracle length, in the sense that*

$$\mathbb{P}\left[|\hat{C}_{n,\alpha}^{\mathrm{sc}}(X_{2n+1})| \leq |C_\alpha^{\mathrm{oracle}}(X_{2n+1})| + \gamma_n\right] \geq 1 - \xi_n,$$

*where $\gamma_n = 4C\eta_n + K\left(\epsilon_n + 2\eta_n^{1/3}\right) \to 0$, and $\xi_n = \delta_n + 2n^{-2} \to 0$.*

*(ii) Asymptotic oracle conditional coverage, in the sense that*

$$\mathbb{P}\left[\mathbb{P}\left[Y \in \hat{C}_{n,\alpha}^{\mathrm{sc}}(X_{2n+1}) \mid X_{2n+1}\right] \geq 1 - \alpha - \epsilon_n\right] \geq 1 - \zeta_n,$$

*where $\epsilon_n = 2/n + 5\eta_n^{1/3} + (1 + 2K)\eta_n + 2\sqrt{(\log n)/n} \to 0$ and $\zeta_n = \eta_n^{1/3} + \eta_n + 2n^{-2} \to 0$.*

***Proof of Theorem S2.*** Assumption 4 (unimodality) and Assumption 5 (smoothness) imply the optimal intervals solving (7) for different values of $\tau$ are nested, so we do not need to define the prediction intervals recursively. More precisely, under Assumptions 4 and 5, $\hat{C}_{n,\alpha}^{\mathrm{sc}}(X_{2n+1}) = \hat{S}(X, \hat{\pi}, \hat{Q}_{1-\alpha}(E_i))$, where $\hat{S}(X, \hat{\pi}, \hat{Q}_{1-\alpha}(E_i))$ is the solution to the optimization problem in (7) with $S^- = \emptyset$ and $S^+ = \{1, \ldots, m\}$, while $\hat{Q}_\tau(E_i)$ is the $\lceil \tau(n+1) \rceil$ smallest value among $\{E_i\}$ for $i \in \{n+1, \ldots, 2n\}$ for any $\tau \in (0, 1]$. The above simplification, combined with the assumed lack of randomization, will simplify our task considerably.

In order to keep the notation consistent, we will refer to $C_\tau^{\mathrm{oracle}}(X_{2n+1})$ as the optimal solution $S^*(X_{2n+1}, f, \tau)$ to the oracle optimization problem in (3)–(4), where $f$ is the conditional probability density of $Y \mid X$. Furthermore, without loss of generality, we divide the conformity scores $E_i$ of Algorithm 1 by $T$, so that they take values between 0 and 1 and can be directly interpreted as probabilities.

The proof will develop as follows.

(i) **Near-optimal length.** First, we will prove in Lemma S1 that each interval $\hat{S}(X, \hat{\pi}, \tau)$ typically cannot be much wider than the corresponding oracle interval $S^*(X, f, \tau + \delta\tau)$, for any fixed $\tau$ and an appropriately small $\delta\tau > 0$. This result will be based on Assumption 2 (consistency). Then, we will prove in Lemma S2 that $\hat{Q}_{1-\alpha}(\{E_i\}_{i \in \mathcal{D}^{\mathrm{cal}}})$ cannot be much larger than $1 - \alpha$; this will be based on Assumption 2 (consistency) as well as on Assumption 1 (i.i.d. data), which makes the empirical quantiles to concentrate around their population values. Combining the above two lemmas will allow us to conclude that $\hat{S}(X, \hat{\pi}, \hat{\tau})$ cannot typically be much wider than $S^*(X, f, 1 - \alpha)$.

(ii) **Near-conditional length.** First, we will prove in Lemma S3 that $\hat{Q}_{1-\alpha}(\{E_i\}_{i \in \mathcal{D}^{\mathrm{cal}}})$ cannot be much smaller than $1 - \alpha$; again, this relies on the concentration of empirical quantiles due to the i.i.d. assumption. Then, we will prove in Lemma S4 that the conditional coverage of $\hat{S}(X, \hat{\pi}, \tau)$ cannot be much smaller than $\tau$, for any fixed $\tau \in (0, 1]$; this result relies on the consistency assumption. Combining the above two lemmas will allow us to conclude that $\hat{S}(X, \hat{\pi}, \hat{\tau})$ cannot typically have conditional coverage much smaller than $1 - \alpha$.

While Assumptions 1–2 will be critical, as previewed above, Assumptions 3–5 will play a subtler yet important role in connecting the various pieces.

**Lemma S1.** *Under Assumptions 1–5, for any $\tau \in (0, 1)$ and $X \perp\!\!\!\perp \mathcal{D}^{\mathrm{train}}$,*

$$\mathbb{P}\left[|\hat{S}(X, \hat{\pi}, \tau)| \leq |S^*(X, f, \tau + 2\eta_n^{1/3})| + 4C\eta_n\right] \geq 1 - \delta_n,$$

*where $\delta_n := \eta_n^{1/3} + \eta_n$.*

**Lemma S2.** *For any $\tau \in (0, 1]$, let $\hat{Q}_\tau(E_i)$ denote the $\lceil \tau(n + 1) \rceil$ smallest value among the conformity scores $\{E_i\}$ for $i \in \mathcal{D}^{\mathrm{cal}}$, where $n = |\mathcal{D}^{\mathrm{cal}}|$ and*

$$E_i := \min\left\{\tau_t \in \{0, 1/T_n, \ldots, (T_n - 1)/T_n, 1\} : Y_i \in \hat{S}(X_i, \hat{\pi}, \tau_t)\right\}.$$

*Then, under Assumptions 1–5, for any $c > 0$,*

$$\mathbb{P}\left[\hat{Q}_\tau(E_i) \leq \tau + \epsilon_n\right] \geq 1 - 2n^{-2c^2},$$

*where $\epsilon_n := 3/n + 3\eta_n^{1/3} + \eta_n + 2c\sqrt{(\log n)/n}$.*

**(i) Near-optimal length.** Define $\delta_n := \eta_n^{1/3} + \eta_n$ as in Lemma S1, and $\epsilon_n := 3/n + 3\eta_n^{1/3} + \eta_n + 2c\sqrt{(\log n)/n}$, for any $c > 0$, as in Lemma S2. In the event that $\hat{Q}_{1-\alpha}(E_i) \leq 1 - \alpha + \epsilon_n$,

$$\mathbb{P}\left[|\hat{S}(X, \hat{\pi}, \hat{Q}_{1-\alpha}(E_i))| \leq |S^*(X, f, 1 - \alpha + \epsilon_n + 2\eta_n^{1/3})| + 4C\eta_n\right]$$
$$\geq \mathbb{P}\left[|\hat{S}(X, \hat{\pi}, 1 - \alpha + \epsilon_n)| \leq |S^*(X, f, 1 - \alpha + \epsilon_n + 2\eta_n^{1/3})| + 4C\eta_n\right]$$
$$\geq 1 - \delta_n,$$

where the second inequality follows by applying Lemma S1 with $\tau = 1 - \alpha + \epsilon_n$. Further, as Lemma S2 tells us the above event occurs with high probability,

$$\mathbb{P}\left[\hat{Q}_{1-\alpha}(E_i) \leq 1 - \alpha + \epsilon_n\right] \geq 1 - 2n^{-2c^2},$$

in general we have that

$$\mathbb{P}\left[|\hat{S}(X, \hat{\pi}, \hat{Q}_{1-\alpha}(E_i))| \leq |S^*(X, f, 1 - \alpha + \epsilon_n + 2\eta_n^{1/3})| + 4C\eta_n\right] \geq 1 - \delta_n - 2n^{-2c^2}.$$

By Assumption 3, $f(y \mid x) > 1/K$ for all $y \in [-C, C]$. This implies $|S^*(X, f, \tau)|$ is $K$-Lipschitz as a function of $\tau$. Therefore,

$$\mathbb{P}\left[|\hat{S}(X, \hat{\pi}, \hat{Q}_{1-\alpha}(E_i))| \leq |S^*(X, f, 1 - \alpha)| + 4C\eta_n + K\left(\epsilon_n + 2\eta_n^{1/3}\right)\right]$$
$$\geq \mathbb{P}\left[|\hat{S}(X, \hat{\pi}, \hat{Q}_{1-\alpha}(E_i))| \leq |S^*(X, f, 1 - \alpha + \epsilon_n + 2\eta_n^{1/3})| + 4C\eta_n\right]$$
$$\geq 1 - \delta_n - 2n^{-2c^2}.$$

Hence we have proved that

$$\mathbb{P}\left[|\hat{S}(X, \hat{\pi}, \hat{Q}_{1-\alpha}(E_i))| \leq |S^*(X, f, 1 - \alpha)| + \gamma_n\right] \geq 1 - \xi_n,$$

where $\gamma_n = 4C\eta_n + K(\epsilon_n + 2\eta_n^{1/3})$ and $\xi_n = \delta_n + 2n^{-2c^2}$. For simplicity, we then set $c = 1$. This completes the proof of (i).

**Lemma S3.** *For any $\tau \in (0, 1]$, let $\hat{Q}_\tau(E_i)$ denote the $\lceil \tau(n+1) \rceil$ smallest value among $\{E_i\}$ for $i \in \mathcal{D}^{\text{cal}}$, where $n = |\mathcal{D}^{\text{cal}}|$ and*

$$E_i := \min\left\{\tau_t \in \{0, 1/T_n, \ldots, (T_n - 1)/T_n, 1\} : Y_i \in \hat{S}(X_i, \hat{\pi}, \tau_t)\right\}.$$

*Then, under Assumptions 1–5, for any $c > 0$,*

$$\mathbb{P}\left[\hat{Q}_\tau(E_i) \geq \tau - \bar{\epsilon}_n\right] \geq 1 - 2n^{-2c^2},$$

*where $\bar{\epsilon}_n := 2/n + 3\eta_n^{1/3} + (1 + 2K)\eta_n + 2c\sqrt{(\log n)/n}$.*

**Lemma S4.** *Consider a test point $(X, Y) \perp\!\!\!\perp \mathcal{D}^{\text{train}}, \mathcal{D}^{\text{cal}}$. $\forall \tau \in (0, 1]$, under Assumptions 1–5,*

$$\mathbb{P}\left[\mathbb{P}\left[Y \in \hat{S}(X, \hat{\pi}, \tau) \mid X\right] \geq \tau - 2\eta_n^{1/3}\right] \geq 1 - \eta_n^{1/3} - \eta_n.$$

**(ii) Near-conditional coverage.** Define $\bar{\epsilon}_n := 2/n + 3\eta_n^{1/3} + (1 + 2K)\eta_n + 2c\sqrt{(\log n)/n}$ as in Lemma S3. Then, focus on the event

$$\mathcal{E} := \left\{\hat{Q}_{1-\alpha}(E_i) \geq 1 - \alpha - \bar{\epsilon}_n\right\}.$$

In this event, for a new test point $(X, Y) \perp\!\!\!\perp \mathcal{D}^{\text{train}}, \mathcal{D}^{\text{cal}}$,

$$\mathbb{P}\left[\mathbb{P}\left[Y \in \hat{S}(X, \hat{\pi}, \hat{Q}_{1-\alpha}(E_i)) \mid X\right] \geq 1 - \alpha - \bar{\epsilon}_n - 2\eta_n^{1/3}\right]$$
$$\geq \mathbb{P}\left[\mathbb{P}\left[Y \in \hat{S}(X, \hat{\pi}, 1 - \alpha - \bar{\epsilon}_n) \mid X\right] \geq 1 - \alpha - \bar{\epsilon}_n - 2\eta_n^{1/3}\right]$$
$$\geq 1 - \eta_n^{1/3} - \eta_n,$$

where the last inequality follows by applying Lemma S4 with $\tau = 1 - \alpha - \bar{\epsilon}_n$. Finally, note that Lemma S3 says the event $\mathcal{E}$ occurs with probability at least $1 - 2n^{-2}$, if we choose $c = 1$. Therefore,

$$\mathbb{P}\left[\mathbb{P}\left[Y \in \hat{S}(X, \hat{\pi}, \hat{Q}_{1-\alpha}(E_i)) \mid X\right] \geq 1 - \alpha - \bar{\epsilon}_n - 2\eta_n^{1/3}\right] \geq 1 - \eta_n^{1/3} - \eta_n - 2n^{-2}.$$

$\square$

**Proofs of technical lemmas**

The proofs of Lemmas S1–S4 will rely on the following additional lemma, which we state here and prove last.

**Lemma S5.** *Define the event $A_n$ as*

$$A_n := \left\{ x : \sup_{j \in \{1,\ldots,m_n\}} |\hat{F}(b_j \mid x) - F(b_j \mid x)| > \eta_n^{1/3} \right\}.$$

*Then, under Assumptions 1–5, for any $X \perp\!\!\!\perp \mathcal{D}^{\mathrm{train}}$,*

$$\mathbb{P}[X \in A_n] \leq \eta_n^{1/3} + \eta_n.$$

*Furthermore, partitioning the calibration data points into*

$$\mathcal{D}^{\mathrm{cal},a} := \{i \in \{n+1,\ldots,2n\} : X_i \in A_n\}, \quad \mathcal{D}^{\mathrm{cal},b} := \{i \in \{n+1,\ldots,2n\} : X_i \in A_n^{\mathrm{c}}\},$$

*we have that, for any constant $c > 0$,*

$$\mathbb{P}\left[|\mathcal{D}^{\mathrm{cal},a}| \geq n\left(\eta_n^{1/3} + \eta_n\right) + c\sqrt{n \log n}\right] \leq n^{-2c^2}.$$

***Proof of Lemma S1.*** Consider the event $A_n$ defined in Lemma S5,

$$A_n := \left\{ x : \sup_{j \in \{1,\ldots,m_n\}} |\hat{F}(b_j \mid x) - F(b_j \mid x)| > \eta_n^{1/3} \right\},$$

and let us restrict our attention to the case in which $X$ belongs to the complement of $A_n$.

Omitting the explicit dependence on $X$ and $\hat{\pi}$, we can write $\hat{S}(X, \hat{\pi}, \tau) = [\hat{j}_1, \hat{j}_2]$, for some $\hat{j}_1, \hat{j}_2 \in \{1,\ldots,m_n\}$ such that $\hat{F}(b_{\hat{j}_2}) - \hat{F}(b_{\hat{j}_1-1}) \geq \tau$. Because we are assuming $X$ belongs to the complement of $A_n$, the triangle inequality implies $F(b_{\hat{j}_2}) - F(b_{\hat{j}_1-1}) \geq \tau - 2\eta_n^{1/3}$. Consider now the oracle interval $S^*(X, f, \tau + 2\eta_n^{1/3})$, which we can write in short as $[l^*, u^*]$, for some $l^*, u^* \in \mathbb{R}$ such that $F(u^*) - F(l^*) \geq \tau + 2\eta_n^{1/3}$. Define now $j_1', j_2' \in \{1,\ldots,m_n\}$ as the indices of the discretized bins immediately below and above $l^*, u^*$, respectively; precisely,

$$j_1' := \max\{j \in \{1,\ldots,m_n\} : b_j < l^*\},$$
$$j_2' := \min\{j \in \{1,\ldots,m_n\} : b_j > u^*\}.$$

This definition implies

$$b_{j_2'} - b_{j_1'} \leq u^* - l^* + 4C/m_n,$$

as each bin has width $2C/m_n$. Furthermore,

$$\begin{aligned}
\hat{F}(b_{j_2'}) - \hat{F}(b_{j_1'}) &\geq \hat{F}(u^*) - \hat{F}(l^*) \\
&\geq F(u^*) - F(l^*) - 2\eta_n^{1/3} \\
&\geq \tau.
\end{aligned}$$

Above, the first inequality follows from the fact that $j_1' < l^*$ and $j_2' > u^*$, the second inequality follows from the assumption that $X$ belongs to the complement of $A_n$, and the third inequality follows directly from the definition of the oracle. The result implies that $[j_1', j_2']$ would be a feasible solution for the discrete optimization problem solved by $\hat{S}(X, \hat{\pi}, \tau)$; therefore, it must be the case that $\hat{j}_2 - \hat{j}_1 \leq j_2' - j_1'$ because $\hat{j}_2 - \hat{j}_1$ is minimal among all feasible solutions to this problem. Therefore, we can conclude that, if $X$ belongs to the complement of $A_n$, then

$$\begin{aligned}
|\hat{S}(X, \hat{\pi}, \tau)| = b_{\hat{j}_2} - b_{\hat{j}_1} &\leq b_{j_2'} - b_{j_1'} \\
&\leq |S^*(X, f, \tau + 2\eta_n^{1/3})| + 4C/m_n.
\end{aligned}$$

Finally, the proof is complete by applying Lemma S5.

$\square$

***Proof of Lemma S2***. Take any $i \in \mathcal{D}^{\mathrm{cal},b}$, where $\mathcal{D}^{\mathrm{cal},b}$ is defined as in Lemma S5:

$$\mathcal{D}^{\mathrm{cal},b} := \{i \in \{n+1, \ldots, 2n\} : X_i \in A_n^{\mathrm{c}}\},$$

where

$$A_n := \left\{ x : \sup_{j \in \{1, \ldots, m_n\}} |\hat{F}(b_j \mid x) - F(b_j \mid x)| > \eta_n^{1/3} \right\}.$$

For any fixed $t \in \{0, \ldots, T_n\}$ and $\tau_t = t/T_n$, omitting the explicit dependence on $X$ and $\hat{\pi}$, we can write $\hat{S}(X, \hat{\pi}, \tau_t) = [\hat{j}_1, \hat{j}_2]$, for some $\hat{j}_1, \hat{j}_2 \in \{1, \ldots, m_n\}$ such that $\hat{F}(b_{\hat{j}_2}) - \hat{F}(b_{\hat{j}_1-1}) \geq \tau_t$. Then, note that

$$\begin{aligned}
\mathbb{P}[E_i \leq \tau_t] &= \mathbb{P}\Big[Y_i \in \hat{S}(X_i, \hat{\pi}, \tau_t)\Big] \\
&= F(b_{\hat{j}_2}) - F(b_{\hat{j}_1-1}) \\
&\geq \hat{F}(b_{\hat{j}_2}) - \hat{F}(b_{\hat{j}_1-1}) - 2\eta_n^{1/3} \\
&\geq \tau_t - 2\eta_n^{1/3}.
\end{aligned}$$

Above, the first inequality follows from the definition of $\mathcal{D}^{\mathrm{cal},b}$. Equivalently, we can rewrite this as

$$\mathbb{P}\Big[E_i > \tau_t + 2\eta_n^{1/3} + \delta_n\Big] \leq 1 - \tau_t - \delta_n,$$

for any $\delta_n > 0$. Now, partition $\mathcal{D}^{\mathrm{cal},b}$ into the following two disjoint subsets:

$$\begin{aligned}
\mathcal{D}^{\mathrm{cal},b1} &:= \{i \in \mathcal{D}^{\mathrm{cal},b} : E_i \leq \tau_t + 2\eta_n^{1/3} + \delta_n\}, \\
\mathcal{D}^{\mathrm{cal},b2} &:= \{i \in \mathcal{D}^{\mathrm{cal},b} : E_i > \tau_t + 2\eta_n^{1/3} + \delta_n\}.
\end{aligned}$$

As in the proof of Lemma S5, we bound $|\mathcal{D}^{\mathrm{cal},b2}|$ with Hoeffding's inequality. For any $i \in \mathcal{D}^{\mathrm{cal}}$, define $\tilde{E}_i = E_i$ if $i \in \mathcal{D}^{\mathrm{cal},b}$ and $E_i = \tau_t$ otherwise. For any $\epsilon > 0$,

$$\begin{aligned}
&\mathbb{P}\Big[|\mathcal{D}^{\mathrm{cal},b2}| \geq n(1 - \tau_t - \delta_n) + \epsilon\Big] \\
&\leq \mathbb{P}\left[\frac{1}{n} \sum_{i \in \mathcal{D}^{\mathrm{cal},b}} \mathbb{1}\Big[\tilde{E}_i > \tau_t + 2\eta_n^{1/3} + \delta_n\Big] \geq \mathbb{P}\Big[E_i > \tau_t + 2\eta_n^{1/3} + \delta_n\Big] + \frac{\epsilon}{n}\right] \\
&= \mathbb{P}\left[\frac{1}{n} \sum_{i=1}^{n} \mathbb{1}\Big[\tilde{E}_i > \tau_t + 2\eta_n^{1/3} + \delta_n\Big] \geq \mathbb{P}\Big[E_i > \tau_t + 2\eta_n^{1/3} + \delta_n\Big] + \frac{\epsilon}{n}\right] \\
&\leq \mathbb{P}\left[\frac{1}{n} \sum_{i=1}^{n} \mathbb{1}\Big[\tilde{E}_i > \tau_t + 2\eta_n^{1/3} + \delta_n\Big] \geq \mathbb{P}\Big[\tilde{E}_i > \tau_t + 2\eta_n^{1/3} + \delta_n\Big] + \frac{\epsilon}{n}\right] \\
&\leq \exp\left(-\frac{2\epsilon^2}{n}\right).
\end{aligned}$$

Therefore, setting $\epsilon = c\sqrt{n \log n}$, for some constant $c > 0$, yields

$$\mathbb{P}\Big[|\mathcal{D}^{\mathrm{cal},b2}| \geq n(1 - \tau_t - \delta_n) + c\sqrt{n \log n}\Big] \leq n^{-2c^2}.$$

As $|\mathcal{D}^{\mathrm{cal},b1}| = n - |\mathcal{D}^{\mathrm{cal},a}| - |\mathcal{D}^{\mathrm{cal},b2}|$, combining the above result with that of Lemma S5 yields:

$$\mathbb{P}\Big[|\mathcal{D}^{\mathrm{cal},b1}| \geq n\tau_t + n\delta_n - n\left(\eta_n^{1/3} + \eta_n\right) - 2c\sqrt{n \log n}\Big] \geq 1 - 2n^{-2c^2}.$$

If we choose $\delta_n = \tau_t/n + \left(\eta_n^{1/3} + \eta_n\right) + 2c\sqrt{(\log n)/n}$, this becomes

$$\mathbb{P}\Big[|\mathcal{D}^{\mathrm{cal},b1}| \geq \tau_t(n+1)\Big] \geq 1 - 2n^{-2c^2},$$

which means

$$\mathbb{P}\Big[\hat{Q}_{\tau_t}(E_i) \leq \tau_t + \tau_t/n + 3\eta_n^{1/3} + \eta_n + 2c\sqrt{(\log n)/n}\Big] \geq 1 - 2n^{-2c^2}.$$

Now, consider any continuous $\tau \in (0, 1]$, and let $t' = \min\{t \in \{0, \ldots, T_n\} : \tau_t \geq \tau\}$. As $\tau_{t'} \geq \tau$, we know $\hat{Q}_{\tau_{t'}}(E_i) \geq \hat{Q}_\tau(E_i)$. Therefore,

$$\mathbb{P}\Big[\hat{Q}_\tau(E_i) \leq \tau_{t'} + \tau_{t'}/n + 3\eta_n^{1/3} + \eta_n + 2c\sqrt{(\log n)/n}\Big]$$
$$\geq \mathbb{P}\Big[\hat{Q}_{\tau_{t'}}(E_i) \leq \tau_{t'} + \tau_{t'}/n + 3\eta_n^{1/3} + \eta_n + 2c\sqrt{(\log n)/n}\Big]$$
$$\geq 1 - 2n^{-2c^2}.$$

However, as $T_n = n$, we also have that $\tau_{t'} \leq \tau + 1/n$. Therefore,

$$\mathbb{P}\Big[\hat{Q}_\tau(E_i) \leq \tau + 1/n + \tau/n + 1/n^2 + 3\eta_n^{1/3} + \eta_n + 2c\sqrt{(\log n)/n}\Big]$$
$$\geq 1 - 2n^{-2c^2}.$$

Finally, we simplify by replacing $1/n + \tau/n + 1/n^2$ with $3/n$, which preserves the inequality.

$\square$

***Proof of Lemma S3***. The proof is similar to that of the analogous upper bound in Lemma S2. Take any $i \in \mathcal{D}^{\mathrm{cal},b}$, where $\mathcal{D}^{\mathrm{cal},b}$ is defined as in Lemma S5:

$$\mathcal{D}^{\mathrm{cal},b} := \{i \in \{n+1, \ldots, 2n\} : X_i \in A_n^{\mathrm{c}}\},$$

with

$$A_n := \left\{x : \sup_{j \in \{1, \ldots, m_n\}} |\hat{F}(b_j \mid x) - F(b_j \mid x)| > \eta_n^{1/3}\right\}.$$

For any $t \in \{0, \ldots, T_n\}$ and $\tau_t = t/T_n$, omitting the explicit dependence on $X$ and $\hat{\pi}$, we can write $\hat{S}(X, \hat{\pi}, \tau_t) = [\hat{j}_1, \hat{j}_2]$, for some $\hat{j}_1, \hat{j}_2 \in \{1, \ldots, m_n\}$ such that $\hat{F}(b_{\hat{j}_2}) - \hat{F}(b_{\hat{j}_1 - 1}) \geq \tau_t$. Then,

$$\mathbb{P}[E_i \leq \tau_t] = \mathbb{P}\Big[Y_i \in \hat{S}(X_i, \hat{\pi}, \tau_t)\Big]$$
$$= F(b_{\hat{j}_2}) - F(b_{\hat{j}_1 - 1})$$
$$\leq \hat{F}(b_{\hat{j}_2}) - \hat{F}(b_{\hat{j}_1 - 1}) + 2\eta_n^{1/3}$$
$$\leq \tau_t + 2K\eta_n + 2\eta_n^{1/3}.$$

Above, the first inequality follows directly from the definition of $\mathcal{D}^{\mathrm{cal},b}$. The second inequality follows from the observation that $\hat{S}(X_i, \hat{\pi}, \tau_t)$ could not be optimal if $\hat{F}(b_{\hat{j}_2}) - \hat{F}(b_{\hat{j}_1 - 1}) \geq \tau_t + 2K\eta_n$ because it would be possible to obtain a shorter feasible interval by removing either the leftmost or the rightmost bin. In fact, each bin $j$ carries an estimated mass $\hat{\pi}_j \leq K\eta_n$, and $\hat{\pi}$ is assumed to be unimodal. Fix any $\delta_n > 0$, and let us rewrite the above result as

$$\mathbb{P}\Big[E_i \leq \tau_t - 2K\eta_n - 2\eta_n^{1/3} - \delta_n\Big] \leq \tau_t + \delta_n.$$

Now, partition $\mathcal{D}^{\mathrm{cal},b}$ into the following two disjoint subsets:

$$\mathcal{D}^{\mathrm{cal},b1} := \{i \in \mathcal{D}^{\mathrm{cal},b} : E_i \leq \tau - 2K\eta_n - 2\eta_n^{1/3} - \delta_n\},$$
$$\mathcal{D}^{\mathrm{cal},b2} := \{i \in \mathcal{D}^{\mathrm{cal},b} : E_i > \tau - 2K\eta_n - 2\eta_n^{1/3} - \delta_n\}.$$

As in the proof of Lemma S2, we will bound $|\mathcal{D}^{\mathrm{cal},b2}|$ with Hoeffding's inequality. For any $i \in \mathcal{D}^{\mathrm{cal}}$, define $\tilde{E}_i = E_i$ if $i \in \mathcal{D}^{\mathrm{cal},b}$ and $E_i = \tau_t$ otherwise. For any $\epsilon > 0$,

$$\mathbb{P}\big[|\mathcal{D}^{\mathrm{cal},b1}| \geq n(1 - \tau_t - \delta_n) + \epsilon\big]$$

$$\leq \mathbb{P}\left[\frac{1}{n}\sum_{i \in \mathcal{D}^{\mathrm{cal},b}} \mathbb{1}\left[\tilde{E}_i \leq \tau_t - 2K\eta_n - 2\eta_n^{1/3} - \delta_n\right] \geq \mathbb{P}\left[E_i \leq \tau_t - 2K\eta_n - 2\eta_n^{1/3} - \delta_n\right] + \frac{\epsilon}{n}\right]$$

$$= \mathbb{P}\left[\frac{1}{n}\sum_{i=1}^{n} \mathbb{1}\left[\tilde{E}_i \leq \tau_t - 2K\eta_n - 2\eta_n^{1/3} - \delta_n\right] \geq \mathbb{P}\left[E_i \leq \tau_t - 2K\eta_n - 2\eta_n^{1/3} - \delta_n\right] + \frac{\epsilon}{n}\right]$$

$$\leq \mathbb{P}\left[\frac{1}{n}\sum_{i=1}^{n} \mathbb{1}\left[\tilde{E}_i \leq \tau_t - 2K\eta_n - 2\eta_n^{1/3} - \delta_n\right] \geq \mathbb{P}\left[\tilde{E}_i \leq \tau_t - 2K\eta_n - 2\eta_n^{1/3} - \delta_n\right] + \frac{\epsilon}{n}\right]$$

$$\leq \exp\left(-\frac{2\epsilon^2}{n}\right).$$

Therefore, setting $\epsilon = c\sqrt{n \log n}$, for some constant $c > 0$, yields

$$\mathbb{P}\left[|\mathcal{D}^{\mathrm{cal},b1}| \geq n(1 - \tau_t - \delta_n) + c\sqrt{n \log n}\right] \leq n^{-2c^2}.$$

As $|\mathcal{D}^{\mathrm{cal},b2}| = n - |\mathcal{D}^{\mathrm{cal},a}| - |\mathcal{D}^{\mathrm{cal},b1}|$, combining the above result with that of Lemma S5 yields:

$$\mathbb{P}\left[|\mathcal{D}^{\mathrm{cal},b2}| \geq n\tau_t + n\delta_n - n\left(\eta_n^{1/3} + \eta_n\right) - 2c\sqrt{n \log n}\right] \geq 1 - 2n^{-2c^2}.$$

If we choose $\delta_n = \tau_t/n + \left(\eta_n^{1/3} + \eta_n\right) + 2c\sqrt{(\log n)/n}$, this becomes

$$\mathbb{P}\left[|\mathcal{D}^{\mathrm{cal},b2}| \geq \tau_t(n+1)\right] \geq 1 - 2n^{-2c^2},$$

which means

$$\mathbb{P}\left[\hat{Q}_{\tau_t}(E_i) \geq \tau_t - \tau_t/n - 3\eta_n^{1/3} - (1 + 2K)\eta_n - 2c\sqrt{(\log n)/n}\right] \geq 1 - 2n^{-2c^2}.$$

Now, consider any continuous $\tau \in (0, 1]$, and let $t' = \max\{t \in \{0, \dots, T_n\} : \tau_t \leq \tau\}$. As $\tau \geq \tau_{t'}$, we know $\hat{Q}_{\tau}(E_i) \geq \hat{Q}_{\tau_{t'}}(E_i)$. Therefore,

$$\mathbb{P}\left[\hat{Q}_{\tau}(E_i) \geq \tau_{t'} - \tau_{t'}/n - 3\eta_n^{1/3} - (1 + 2K)\eta_n - 2c\sqrt{(\log n)/n}\right]$$

$$\geq \mathbb{P}\left[\hat{Q}_{\tau_t}(E_i) \geq \tau_t - \tau_t/n - 3\eta_n^{1/3} - (1 + 2K)\eta_n - 2c\sqrt{(\log n)/n}\right]$$

$$\geq 1 - 2n^{-2c^2}.$$

However, as $T_n = n$, we also have that $\tau_{t'} \geq \tau - 1/n$. Therefore,

$$1 - 2n^{-2c^2} \leq \mathbb{P}\left[\hat{Q}_{\tau}(E_i) \geq (\tau - 1/n)(1 - 1/n) - 3\eta_n^{1/3} - (1 + 2K)\eta_n - 2c\sqrt{(\log n)/n}\right]$$

$$\leq \mathbb{P}\left[\hat{Q}_{\tau}(E_i) \geq \tau - \tau/n - 1/n + 1/n^2 - 3\eta_n^{1/3} - (1 + 2K)\eta_n - 2c\sqrt{(\log n)/n}\right]$$

Finally, we simplify by replacing $-1/n - \tau/n + 1/n^2$ with $-2/n$, which preserves the inequality.

$\square$

***Proof of Lemma S4.*** Let us begin by conditioning on $X = x$, assuming $x \in A_n^{\mathrm{c}}$, where $A_n$ is defined as in Lemma S5:

$$A_n := \left\{x : \sup_{j \in \{1, \dots, m_n\}} |\hat{F}(b_j \mid x) - F(b_j \mid x)| > \eta_n^{1/3}\right\}.$$

Omitting the explicit dependence on $x$ and $\hat{\pi}$, we can write $\hat{S}(x, \hat{\pi}, \tau) = [\hat{j}_1, \hat{j}_2]$, for some $\hat{j}_1, \hat{j}_2 \in \{1, \ldots, m_n\}$ such that $\hat{F}(b_{\hat{j}_2}) - \hat{F}(b_{\hat{j}_1 - 1}) \geq \tau$.

$$
\begin{aligned}
\mathbb{P}\Big[ Y \in \hat{S}(x, \hat{\pi}, \tau) \mid X = x \Big] &= F(b_{\hat{j}_2}) - F(b_{\hat{j}_1 - 1}) \\
&\geq \hat{F}(b_{\hat{j}_2}) - \hat{F}(b_{\hat{j}_1 - 1}) - 2\eta_n^{1/3} \\
&\geq \tau - 2\eta_n^{1/3},
\end{aligned}
$$

where the second inequality follows from the definition of the complement of $A_n$. Finally, we know from Lemma S5 that $\mathbb{P}[X \in A_n^c] \geq 1 - \eta_n^{1/3} - \eta_n$. $\qquad \square$

***Proof of Lemma S5.*** The first part of this result follows from the definition of $A_n$ by a union bound. For any fixed $j \in \{1, \ldots, m_n\}$,

$$
\begin{aligned}
\mathbb{P}[X \in A_n] = \mathbb{P}\Bigg[ \sup_{j' \in \{1, \ldots, m_n\}} |\hat{F}(b_{j'} \mid x) - F(b_{j'} \mid x)|^2 > \eta_n^{2/3} \Bigg] \\
\leq m_n \, \mathbb{P}\Big[ |\hat{F}(b_j \mid x) - F(b_j \mid x)|^2 > \eta_n^{2/3} \Big] \\
\leq m_n \left( \eta_n^{-2/3} \mathbb{E}\Big[ \mathbb{E}\big[ |\hat{F}(b_j \mid x) - F(b_j \mid x)|^2 \mid \mathcal{D}^{\text{train}} \big] \Big] \right) \\
\leq m_n \left( \eta_n^{-2/3} \eta_n^2 + \eta_n^2 \right) \\
\leq m_n \left( \eta_n^{4/3} + \eta_n^2 \right) \\
\leq \eta_n^{1/3} + \eta_n.
\end{aligned}
$$

The second inequality above is Markov's inequality, while the third inequality follows directly from Assumption 2. The last inequality is a consequence of $m_n = \lfloor \eta_n^{-1} \rfloor$, also from Assumption 2.

The second part of this result follows from Hoeffding's inequality. As we know from the above that $\mathbb{P}[X \in A_n] \leq \eta_n^{1/3} + \eta_n$, for any $\epsilon > 0$,

$$
\begin{aligned}
\mathbb{P}\Big[ |\mathcal{D}^{\text{cal},a}| \geq n \left( \eta_n^{1/3} + \eta_n \right) + \epsilon \Big] &\leq \mathbb{P}\Big[ |\mathcal{D}^{\text{cal},a}| \geq n\mathbb{P}[X \in A_n] + \epsilon \Big] \\
&\leq \mathbb{P}\Bigg[ \frac{1}{n} \sum_{i=n+1}^{2n} \mathbb{1}\,[X_i \in A_n] \geq \mathbb{P}[X_i \in A_n] + \frac{\epsilon}{n} \Bigg] \\
&\leq \exp\left( -\frac{2\epsilon^2}{n} \right).
\end{aligned}
$$

Therefore, setting $\epsilon = c\sqrt{n \log n}$, for some constant $c > 0$, yields

$$
\mathbb{P}\Big[ |\mathcal{D}^{\text{cal},a}| \geq n \left( \eta_n^{1/3} + \eta_n \right) + c\sqrt{n \log n} \Big] \leq n^{-2c^2}.
$$

$\qquad \square$

## S3 Numerical experiments

This section provides additional details about the numerical experiments with synthetic and real data.

### S3.1 Base machine learning models

We estimate the distribution of $Y \mid X$ using the following quantile regression models.

- **Deep neural network**. The network is composed of three fully connected layers with a hidden dimension of 64, and ReLU activation functions. We use the pinball loss [12] to estimate the conditional quantiles, with a dropout regularization of rate 0.1. The network is optimized using Adam [5] with a learning rate equal to 0.0005. We tune the optimal number of epochs by cross validation, minimizing the loss function on the hold-out data points; the maximal number of epochs is set to 2000.

- **Random forest**. We use the Python `Scikit-garden` implementation of quantile regression forests [7]. We adopt the default hyper-parameters, except for the minimum number of samples required to split an internal node, which we set to 50, and the total number of trees, which we fix equal to 100.

Our numerical experiments were conducted on Xeon-2640 CPUs in a computing cluster. Each data set was analyzed using a single core and less than 5 GB of memory; the longest job took less than 12 hours. The computational cost of the novel part of CHR is negligible: the majority of the computing resources were dedicated to training the base models.

## S3.2 Additional experiments with synthetic data

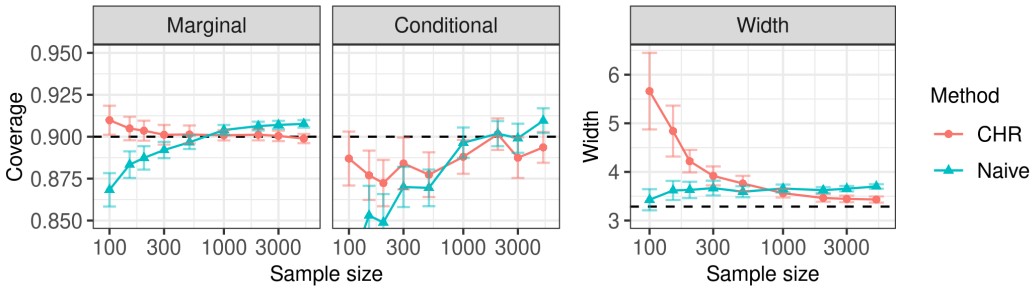

Figure S1: Performance of our method (CHR) compared to that of naive uncalibrated prediction intervals based on the same deep neural network regression model, in the experiments of Figure 3.

## S3.3 Additional experiments with real data

In Section 4.3 of the main article, we compared the performance of our method to that of several benchmarks using a deep neural network base model. Figure S2 provides additional comparisons using a random forest base model. The bottom panel of this figure shows the average interval length. Our method (CHR) significantly outperforms all benchmarks by this metric, as it consistently constructs shorter intervals. The top panel of Figure S2 compares these alternative methods in terms of their worst-slab conditional coverage [2], which we estimate as in [10]. All methods achieve high conditional coverage on most data sets, except for CHR which tends to slightly undercover in the case of the two Facebook data sets (fb1 and fb2). Lastly, we note that all methods achieve exact 90% marginal coverage, as guaranteed theoretically; see Table S1 for additional performance details.

Table S1: Performance of our method and benchmarks on several real data sets, using either a deep neural network or a random forest base model. The numerical values indicate values averaged over 100 random test sets (standard deviations are in parenthesis). Other details are as in Figures 4 and S2.

| | | Neural Network | | | Random Forest | | |
|---|---|---|---|---|---|---|---|
| | | Coverage | | | Coverage | | |
| Data | Method | Marginal | Condit. | Width | Marginal | Condit. | Width |
| bio | CHR | 0.90 (0.01) | 0.88 (0.03) | 13.1 (0.3) | 0.90 (0.01) | 0.90 (0.03) | 10.4 (0.3) |
| | CQR | 0.90 (0.01) | 0.88 (0.03) | 14.5 (0.2) | 0.90 (0.01) | 0.89 (0.03) | 12.9 (0.1) |
| | DCP | 0.90 (0.01) | 0.88 (0.03) | 14.6 (0.3) | 0.90 (0.01) | 0.90 (0.02) | 11.7 (0.2) |
| | DCP-CQR | 0.90 (0.01) | 0.87 (0.03) | 14.8 (0.4) | 0.90 (0.01) | 0.89 (0.03) | 11.9 (0.3) |
| | DistSplit | 0.90 (0.01) | 0.88 (0.03) | 14.7 (0.3) | 0.90 (0.01) | 0.90 (0.03) | 11.9 (0.3) |
| blog | CHR | 0.90 (0.01) | 0.88 (0.03) | 10.9 (1.2) | 0.90 (0.01) | 0.88 (0.03) | 10.3 (1.2) |
| | CQR | 0.90 (0.01) | 0.87 (0.04) | 15.0 (1.5) | 0.90 (0.01) | 0.90 (0.02) | 21.1 (1.6) |
| | DCP | 0.90 (0.01) | 0.89 (0.03) | 1422.3 (0.1) | 0.90 (0.01) | 0.90 (0.03) | 1421.3 (0.1) |
| | DCP-CQR | 0.90 (0.01) | 0.86 (0.04) | 14.0 (1.4) | 0.90 (0.01) | 0.90 (0.03) | 21.4 (1.8) |
| | DistSplit | 0.90 (0.01) | 0.87 (0.04) | 15.8 (1.6) | 0.90 (0.01) | 0.89 (0.03) | 16.7 (1.8) |
| fb1 | CHR | 0.90 (0.01) | 0.87 (0.04) | 10.6 (0.9) | 0.90 (0.01) | 0.87 (0.04) | 11.2 (0.9) |
| | CQR | 0.90 (0.01) | 0.89 (0.03) | 14.6 (1.0) | 0.90 (0.01) | 0.90 (0.02) | 19.2 (1.5) |
| | DCP | 0.90 (0.01) | 0.90 (0.03) | 1303.3 (0.1) | 0.90 (0.01) | 0.90 (0.03) | 1302.6 (0.1) |
| | DCP-CQR | 0.90 (0.01) | 0.89 (0.03) | 13.2 (1.1) | 0.90 (0.01) | 0.90 (0.03) | 19.4 (1.7) |
| | DistSplit | 0.90 (0.01) | 0.89 (0.03) | 14.3 (1.1) | 0.90 (0.01) | 0.90 (0.03) | 16.5 (1.3) |
| fb2 | CHR | 0.90 (0.01) | 0.87 (0.03) | 11.0 (0.9) | 0.90 (0.01) | 0.86 (0.03) | 10.8 (0.9) |
| | CQR | 0.90 (0.01) | 0.89 (0.03) | 14.2 (0.9) | 0.90 (0.01) | 0.90 (0.03) | 17.7 (1.4) |
| | DCP | 0.90 (0.01) | 0.90 (0.03) | 1964.0 (0.1) | 0.90 (0.01) | 0.89 (0.03) | 1963.4 (0.1) |
| | DCP-CQR | 0.90 (0.01) | 0.89 (0.03) | 12.8 (1.1) | 0.90 (0.01) | 0.89 (0.03) | 17.8 (1.6) |
| | DistSplit | 0.90 (0.01) | 0.89 (0.03) | 14.2 (1.1) | 0.90 (0.01) | 0.89 (0.03) | 15.1 (1.3) |
| meps19 | CHR | 0.90 (0.01) | 0.90 (0.02) | 20.1 (1.3) | 0.90 (0.01) | 0.89 (0.03) | 18.4 (1.3) |
| | CQR | 0.90 (0.01) | 0.89 (0.03) | 29.3 (1.2) | 0.90 (0.01) | 0.90 (0.02) | 32.6 (1.3) |
| | DCP | 0.90 (0.01) | 0.89 (0.03) | 559.3 (0.0) | 0.90 (0.01) | 0.89 (0.03) | 559.0 (0.0) |
| | DCP-CQR | 0.90 (0.01) | 0.89 (0.03) | 33.3 (2.3) | 0.90 (0.01) | 0.90 (0.03) | 32.2 (2.0) |
| | DistSplit | 0.90 (0.01) | 0.90 (0.03) | 30.0 (2.3) | 0.90 (0.01) | 0.90 (0.03) | 29.8 (2.2) |
| meps20 | CHR | 0.90 (0.01) | 0.90 (0.02) | 19.1 (1.2) | 0.90 (0.01) | 0.90 (0.02) | 17.7 (1.1) |
| | CQR | 0.90 (0.01) | 0.88 (0.02) | 28.1 (1.0) | 0.90 (0.01) | 0.90 (0.03) | 30.5 (1.3) |
| | DCP | 0.90 (0.01) | 0.89 (0.03) | 520.3 (0.0) | 0.90 (0.01) | 0.89 (0.03) | 520.1 (0.0) |
| | DCP-CQR | 0.90 (0.01) | 0.89 (0.02) | 32.1 (2.2) | 0.90 (0.01) | 0.90 (0.02) | 29.9 (2.0) |
| | DistSplit | 0.90 (0.01) | 0.89 (0.03) | 28.8 (2.0) | 0.90 (0.01) | 0.90 (0.03) | 27.9 (2.0) |
| meps21 | CHR | 0.90 (0.01) | 0.90 (0.03) | 20.5 (1.2) | 0.90 (0.01) | 0.90 (0.03) | 19.2 (1.1) |
| | CQR | 0.90 (0.01) | 0.89 (0.03) | 30.1 (1.3) | 0.90 (0.01) | 0.90 (0.02) | 33.4 (1.4) |
| | DCP | 0.90 (0.01) | 0.89 (0.03) | 531.3 (0.0) | 0.90 (0.01) | 0.89 (0.03) | 531.0 (0.0) |
| | DCP-CQR | 0.90 (0.01) | 0.89 (0.03) | 34.5 (2.4) | 0.90 (0.01) | 0.90 (0.02) | 32.9 (2.1) |
| | DistSplit | 0.90 (0.01) | 0.90 (0.03) | 30.5 (2.0) | 0.90 (0.01) | 0.90 (0.03) | 30.6 (2.1) |

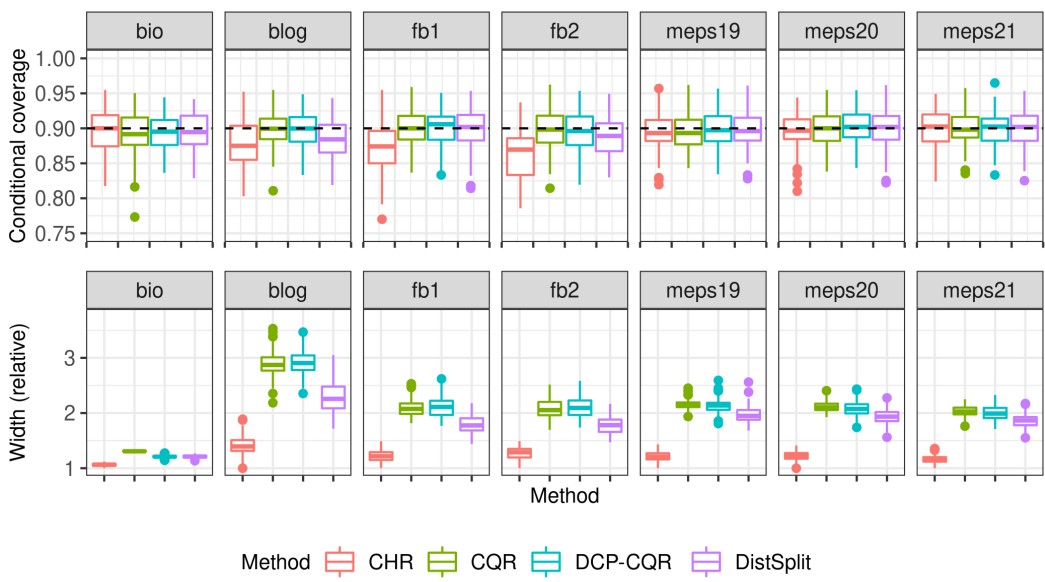

Figure S2: Performance of our method and benchmarks on several real data sets, using a random forest base model. Other details are as in Figure 4.

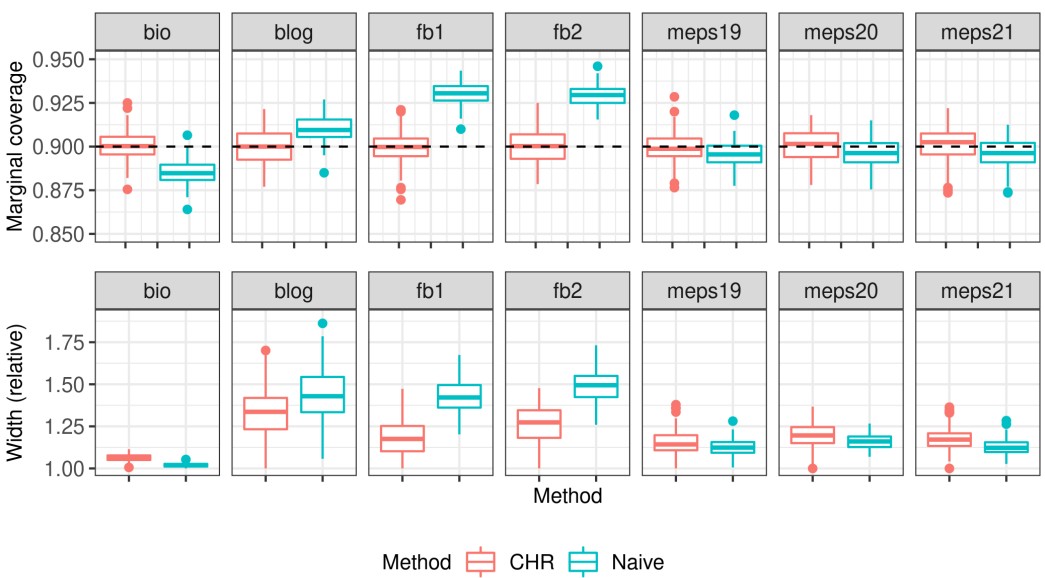

Figure S3: Performance of our method (CHR) compared to that of naive uncalibrated prediction intervals based on the same deep neural network regression model, in the experiments with real data of Figure 4. Note that the top part of this plot shows marginal coverage.