# OpenReview forum: "Conformal Prediction using Conditional Histograms"
_NeurIPS.cc/2021/Conference — NeurIPS 2021 Spotlight_

### Official Review · Reviewer_ucSx · 2021-07-04

**Rating:** 9
**Confidence:** 5

**Summary:**

The paper shows how to obtain *short* prediction intervals that approximately have the right conditional coverage. In order to do so, the method estimates the conditional distribution of a new label given its features, and then uses a conformal score based on the quantiles implied by such estimate to create the prediction intervals. The experiments show that the method outperforms other quantile-based conformal methods in terms of width while still approximately controlling conditional coverage. Some theoretical results that prove converge to the oracle (i.e., the shortest prediction interval) are also shown.

**Limitations And Societal Impact:**

Limitations have been properly addressed.

**Main Review:**

The paper is well written and technically sound. Given the importance of accurate uncertainty quantification methods, the paper is also very relevant.

As far as I know, this is the first paper on conformal methods that attempt to find *short* prediction *intervals*; most conformal-quantile approaches are based on intervals that have the form $(q_\alpha,q_{1-\alpha})$, which are only the shortest intervals if the underlying distribution is symmetric. Thus, the paper has an important and novel goal.

As the authors mention, there has however been work on how to recover short prediction *regions* (which may not be intervals). Although such regions will be smaller than the oracle intervals if the underlying distribution is **not** unimodal, the fact that the CHR yields intervals is an advantage in some settings (for instance, they are easier to report). This advantage should be emphasized.

Related to this point, the conformity score (Eq. 12) is essentially (up to the randomization need due to the discrete nature of the estimated density) the HPD-split score introduced by [20] (https://arxiv.org/abs/2007.12778). Indeed, the score is (a monotonic transformation of) the area under a region with high density. The main difference is that CHR has an additional constrain that the output regions must be intervals. Indeed, Theorem 2 is very similar to Theorem 22 of [20], but apparently, the HPD-split method does not need to satisfy Assumption 4 (unimodality) to recover the oracle region. Thus, in principle, HPD-split could give even smaller regions (as they don't need to be intervals), while still approximately controlling conditional coverage. It would be interesting to see how it compares to CHR in unimodal settings.

It also seems to me that the conformity score in (12) could actually be computed using any conditional density estimator (although Eq. 7 would have to be slightly generalized to mimick Eq. 4). That is, there is no need for the method to be restricted to the histogram-based approach (although such an approach is interesting). In other words, the CHR score is agnostic to the density estimator being used, which is an attractive advantage.

Minor comment: It is not clear to me how conditional coverage was computed/estimated on real datasets. It doesn't seem to be trivial, because it is a conditional probability per se.

**Time Spent Reviewing:**

10

---

> ### Author Response · Authors · 2021-08-09
> **Response to reviewer ucSx**
>
> We are grateful to reviewer ucSx for the thoughtful review and the very encouraging feedback. We feel flattered by such a high score.
>
> As correctly pointed out by this reviewer, the approach taken by our method bears some similarity to that of the HPD-split score introduced by [20], but feel it's worth emphasizing more why the novelty which distinguishes our method (the explicit computation of short prediction *intervals*) is both practically relevant and methodologically significant. Our submitted manuscript already attempts to make clear the distinction between our work and that of [20], but in light of this comment we realize that perhaps even more space should be dedicated to explaining this important point and we would be happy to do so if given the opportunity to do a minor revision. Please note that the rest of this answer closely resembles that to a related comment by reviewer prSn, which we report here for completeness.
>
> First, while we certainly do not claim they are universally preferable, intervals are often naturally much more interpretable than arbitrary regions. Further, reporting a non-convex prediction region conveys a level of statistical confidence that may be hard to justify.
> For example, if we were to report to a physician that the likely blood pressure or a patient with certain characteristics will be, at some point in the future, within  [120,129] mm Hg, we are likely to be of help. However, a statistician who reports to the physician that the patient's future blood pressure is likely to be within the following region, ( [120, 120.012] U [120.015, 120.05] U [121, 122.7] U [123.1, 127.2] U [127.8, 129] ) mm Hg, would risk being taken less seriously. Indeed, it would not be clear (a) whether the multi-modality is significant or a spurious consequence of overfitting, and (b) how the physician would act upon this prediction any differently than if it had been  [120,129] mm Hg. As thoughtfully suggested by this reviewer, the advantage of prediction intervals over sets could be emphasized more explicitly. We would be happy to do so if given the opportunity to do a minor revision.
>
> Of course, one can always transform a one-dimensional non-convex predictive region into an interval by taking its convex hull. However, if we know from the beginning that we wish to report an interval, why take such an indirect route? Certainly in that case it would seem more intuitive and statistically sound to apply a method that is explicitly designed to find the shortest predictive intervals with valid coverage, rather than to resort to heuristic patchworks of different approaches.
>
> This important conceptual difference between our work and HPD-split in [20] is at the heart of the reason why we did not explicitly compare our method to the latter in simulation where the data distribution is unimodal. First, our method is already guaranteed to perform better at this task (under the idealized asymptotic regime of our theoretical analysis, of course). Second, this comparison would be unfair to [20] because HPD-split is specifically designed to be applied in multi-modal settings, not to report prediction intervals. Third, we already compare to the state-of-the-art method designed to report prediction intervals, one of which was developed in [21] by the very same authors of [20].
>
> Regarding the flexibility of the conformity scores in (12), it is true that it could be applied using any conditional density estimator. Indeed, it would be more accurate to say that the expression in (12) indicates a general recipe for computing relatively adaptive conformity scores rather than fully describing the specific ones we adopt in this paper. This type of conformity scores was originally proposed in a different paper [31] within the context of multi-class classification, as mentioned at the beginning of the related work section. The potential flexibility of this family of conformity scores was already noted in [31], but that work did not explore their use beyond the classification setting. The contribution of the present paper is to develop a practical implementation of such adaptive conformity scores that leads to short predictive intervals for regression. We thought we had already given credit in Section 1.3 (related work) to [31] for inspiring our conformity scores for regression, but if the reviewers deems it necessary we would be happy to refer to [31] again in Section 2.
>
> Regarding the definition of the empirical conditional coverage, this is defined as the worst-slab conditional coverage, first proposed in [9]. However, our practical implementation of the worst-slab conditional coverage is slightly different from that of [9], as we utilize the code of [31] which differs insofar as it uses data-splitting to avoid the negative estimation bias which may otherwise result from the original implementation of [9]. This is currently mentioned in Section 4.2. Thanks to this reviewer's comment, we realize many readers may not yet be familiar with the definition and estimation of worst-slab conditional coverage, so we would be happy to recall them in a little more detail if given the opportunity to do a minor revision.

---

> > ### Comment · Reviewer_ucSx · 2021-09-10
> > **Thanks**
> >
> > Thank you for the clarifications; they were very helpful!

---

### Official Review · Reviewer_QWwn · 2021-07-12

**Rating:** 7
**Confidence:** 3

**Summary:**

This paper introduces a conformity score that aims to decrease average interval lengths and improve conditional coverage from a black-box estimate  $\hat{P}(y \mid x)$ for split conformal prediction. The method, conformal histogram regression (CHR), involves first binning the space of $Y$, resulting in a conditional histogram from which approximate oracle intervals can be computed. A nested sequence of these intervals is then created for a sequence of predictive miscoverage values $\tau$, where $\tau$ will be close to $\alpha$ if $\hat{P}(y \mid x)$ is a good estimate. The value of $\tau$ is selected through a conformity score, and the authors show their method obtains finite marginal coverage and asymptotic conditional coverage. The authors then demonstrate the method on a few examples.

**Limitations And Societal Impact:**

The authors have described the limitations of their method - in particular their method does not control for upper and lower miscoverage, and they provide alternative recommendations.

**Main Review:**

Strengths:

This paper is an interesting natural extension of [Romano et al., 2020] from discrete to continuous $Y$, and also connects to the nested conformal interpretation of [Gupta et al., 2019]. Although the conformity score has been previously introduced, the construction of the nested prediction sets through histograms is novel. The asymptotic results are also stronger than previous works, where both asymptotic oracle length and conditional coverage are attained. The empirical results demonstrate superior conditional coverage and average interval lengths to other methods, especially for skewed data. The paper is well-written and enjoyable to read.

#####################################################################

Weaknesses/Questions:

I only have a few minor points:

1.) For equation (7), does treating $|u-l|$ as the length require the bins to be equally spaced? I don't think this is stated.

2.) It may be good to briefly mention the negligible computational cost of CHR (which is in the appendix) in the main paper to help motivate the method. A rough example of some run-times in the experiments may also be useful for readers looking to apply the method.

3.) Just a few typographical/communication points:
- I found Section 2.2 slightly difficult to read, as the notation gets a little heavy. This may not be necessary, but the authors could consider presenting the nested intervals without randomization (e.g. after Line 119), with the randomization in the Appendix, as it is not needed in Theorem 2. This would give more room for intuitive discussions, related to my next point.
- It may be helpful to introduce some intuition on the conformity score in equation (12) and why we need the sets to be nested for readers unfamiliar with previous work, perhaps at the start of Section 2.3.
- Line 113: $\epsilon$ is mentioned here before it is defined
- Line 188: 'increased' instead of 'increase'

#####################################################################

Overall:

This paper is an interesting extension of previous work, and the provided asymptotic justifications of attaining oracle width and conditional coverage is useful. The method is also general and can empirically provide better average widths and conditional coverage than other methods, particularly under skewed data, making it useful in practice.

#####################################################################

References:

Romano, Y., Sesia, M., \& Candes, E. (2020). Classification with Valid and Adaptive Coverage. Advances in Neural Information Processing Systems, 33, 3581-3591.

Gupta, C., Kuchibhotla, A. K., \& Ramdas, A. K. (2019). Nested conformal prediction and quantile out-of-bag ensemble methods. arXiv preprint arXiv:1910.10562.

**Time Spent Reviewing:**

5 hours

---

> ### Author Response · Authors · 2021-08-09
> **Response to reviewer QWwn**
>
> We are grateful to reviewer QWwn for the thoughtful review and the encouraging feedback.
>
> 1) Yes, treating |u-l| as the length requires the bins to be equally spaced. We apologize for forgetting to mention this explicitly, and we will certainly rectify this omission if given the opportunity to do a minor revision. Of course, there is no real need for our method to utilize equally spaced bins; this choice is simply due to notational simplicity. We will also make sure this point is clear in the revision. Thank you for bringing this issue to our attention!
>
> 2) We agree that it may be good to briefly mention the negligible computational cost of CHR in the main paper. We will do so if given the opportunity to do a minor revision.
>
> 3) Typographical/communication points
>
>    - We agree that the exposition may benefit if we defer the randomization to the appendix, as that will simplify the notation without much loss of essential content. We will do so if given the opportunity to do a minor revision.  This was also suggested by reviewer prSn.
>    - As mentioned in the answer to the comments of reviewer ucSx, the conformity scores in (12) are motivated by the related work of [31] within the context of multi-class classification. This is currently mentioned in Section 1.3 (related work), but we agree it may be a good idea to repeat it, and perhaps expand upon it, in Section 2.3. We will do so if given the opportunity to do a minor revision.
>    - Again, we agree with the reviewer and gladly accept the suggestion. It would be clearer if we defined ε before line 113.
>    - Thank you for pointing out the typo on line 188.

---

> > ### Comment · Reviewer_QWwn · 2021-08-23
> > **Reply to Rebuttal**
> >
> > Thank you for the helpful clarifications!

---

### Official Review · Reviewer_mmq9 · 2021-07-15

**Rating:** 7
**Confidence:** 4

**Summary:**

This paper proposes an extension to conformal prediction that adapts to skewed data, and can achieve better conditional coverage (and provably achieves conditional coverage asymptotically). Conformal prediction in general is a methodology for constructing confidence sets that output likely response candidates $\widehat{\mathcal{C}}(X) \subseteq \mathcal{Y}$ for and input $X$, rather than a single value. The goal is to ensure that $\widehat{\mathcal{C}}$ covers the true response variable, $Y$, with specifiably high probability. While many prior methods provably control *marginal* coverage, *conditional* coverage is a much harder (and more practically important) goal (albeit impossible in finite samples in the general case). This paper attacks this problem by developing a novel conformalization strategy that leverages calibrated estimates of the conditional density $Y \mid X$ to obtain approximate conditional coverage in finite samples (empirically), and asymptotic conditional coverage (theoretically).

**Limitations And Societal Impact:**

I don't see any potential negative societal impacts other than those inherited by the underlying wrapped algorithm this method gives confidence intervals for. Conditional coverage is overall more desirable than marginal coverage in terms of mitigating risks like fairness.

**Main Review:**

=== Strengths ===

- The motivation of the paper is very strong; achieving the harder task of approximate conditional coverage is far more important than marginal coverage for most practical problems.

- The writing for the most part is fairly clear, with few exceptions (see questions below).

- The theoretical analysis is quite comprehensive, with more precise guarantees than several other asymptotically conditional conformal methods.

- Empirically the proposed method performs well relative to baselines.

=== Weaknesses ===

- Using estimates of the conditional distribution have been proposed before, most relevantly perhaps in CD-Split (cited) and HPD-Split (also by Izbicki et. al., 2021). Dist-Split is compared to in this work, though not the other methods (although CD-Split is designed for multi-model targets which are assumed to not exist here, it reportedly seems to work empirically better than Dist-Split even across unimodal tasks). Due to the similarities in approaches, it would be helpful to give better intuition as to why the proposed histogram-based method works better. The comments on line 192 are also confusing in terms of what exactly they mean, as Dist-Split in Izbicki et. al., 2020 has similar guarantees (Thm. 2.5 and Corollary 2.6). More discussion on this would be helpful.

- Achieving conditional coverage relies on the consistency of the histogram-based conditional estimator, which scales linearly with the number of bins (which goes to infinity). It seems that achieving conditional coverage as the number of data points grows is at odds with computational complexity, as opposed to a potentially more scalable density estimator (either way, both need to be calibrated).

- The quality of the conditional estimator will likely degrade considerably in high dimensional problems. The tasks considered in this work are also fairly simple. It would be good to see results on more compelling tasks; i.e. that would illustrate the value of the proposed extension across tasks relevant to the NeurIPS community.

=== Questions ===

- Section 2.2 is a bit dense to follow at first in terms of motivation, and it's not entirely clear what the motivation is for constructing the nested sets based off of the initial $\bar{t}$. Instead of conformalizing the index of the set per Eq. (12), what is the downside to conformalizing $\tau$ directly? I.e., one would find the smallest $\tau$ such that $(1 - \alpha)\frac{1 + |\mathcal{D}^{\mathrm{cal}}|}{|\mathcal{D}^{\mathrm{cal}}|}$ fraction of calibration instances satisfy $y \in \underset{(l, u), l \leq u}{\arg\\!\min} \\{ l - u  \colon \sum_{j=l}^{u} \pi_j(x) \geq \tau \\}$. It seems that this is due to the randomization strategy in Eq. (8), which should produce slightly more efficient intervals depending on how coarse the binning is?

=== Minor ===

- Typo, line 188 "increased flexibility"
- Supplement, line 16, I believe should have $\hat{c}$ instead of $c$ in the numerator.

=== Justification for score ===

In general, this paper is a valuable contribution to the growing body of approaches for approximate conditional conformal inference. There are a couple of clarity/differentiation concerns that could be addressed in the writing, and the method is similar to several already existing approaches (e.g., extending Romano et. al., 2020 to a regression setting, including the binning strategy as compared to Izbicki et. al., 2020)---but overall the work is quite solid.

=== Citations ===
- Izbicki et. al., 2020. Flexible distribution-free conditional predictive bands using density estimators.
- Izbicki et. al., 2021. CD-split and HPD-split: efficient conformal regions in high dimensions.
- Romano et. al., 2020. Classification with Valid and Adaptive Coverage.

**Time Spent Reviewing:**

8

---

> ### Author Response · Authors · 2021-08-09
> **Response to reviewer mmq9**
>
> We are grateful to reviewer mmq9 for the thoughtful review and the encouraging feedback.
>
> Weaknesses
>
>  - Thank you for pointing out that we forgot to mention the HPD-Split method explicitly. Indeed, we accidentally cited the first version of [20] instead of the second one in which HPD-Split is presented. We will rectify this imprecision if given the opportunity to do a minor revision. However, we should also emphasize that HPD-Split is similar to CD-Split in spirit, as both methods are specifically designed to compute non-convex predictive regions for multi-model data [20], not short predictive intervals.
>  Please note that related comments were also raised by reviewers prSn and ucSx; see our answers to those reviewers for a more detailed discussion of why we care about prediction intervals.
>   We must disagree with this reviewer regarding the theoretical guarantees of Dist-Split [21], which is the method closest to ours insofar as it also seeks short predictive intervals. It is not true that the theoretical guarantees of Dist-Split are the same as ours. As correctly mentioned by the reviewer, Thm. 2.5 and Corollary 2.6 in [21], establish that Dist-Split [21] is asymptotically equivalent to an oracle, under suitable consistency assumptions, in a sense that is analogous to that of our result. However, there is a fundamental difference between our oracle and theirs! Our oracle is able to compute the shortest prediction intervals with the desired conditional coverage, while theirs computes the shortest prediction intervals *among those with pre-specified upper and conditional miscoverage rates*. These two oracles can be very different if the data distribution is not symmetric, as shown intuitively in Figure 1 and demonstrated by the increased empirical performance of our method. We have noted that other reviewers appreciated this point, but we would nonetheless be happy to spend some additional words to explain it even more clearly if given the opportunity to do a minor revision. Thank you for suggesting to emphasize this!
>
>  - We are not sure that we understand what the reviewer means by "a more scalable density estimator". Density estimation is a much more difficult statistical task than estimating the heights of finitely many histogram bins. Indeed, a non-parametric density estimator is essentially a histogram with an infinite number of bins. Of course, the problem would become simpler if one were willing to introduce parametric assumptions, but this would go beyond the scope of this paper, which focuses on non-parametric methods. Regardless, even if one wanted to start from a smooth conditional density estimator, we would argue that the construction of the shortest predictive intervals with guaranteed marginal coverage should still take place using the conformal sequence of nested sets described in our paper. Indeed, we make it clear that our approach to density estimation through multi-quantile regression is a convenient and effective practical solution, but it is by no means unique.
>
>  - The reviewer is correct about the first point in this comment: estimating conditional distributions becomes more challenging in high dimensions. However, this is a fundamental difficulty of high-dimensional regression, not merely a limitation of our method. First, increasing dimensions also make the problem harder for all other non-parametric conformal regression methods based on flexible machine-learning models, since in one way or another they also require estimating some functional of the conditional distribution, either through quantile regression or other density estimation approaches, for example. Second, while our method is based on a conditional histogram that reminds us of classical density estimation, our performance is not directly measured in terms of how closely the histogram resembles the true density. Instead, loosely speaking, our method only needs to accurately detect how the response values are skewed and then estimate reasonably well the lower and upper conditional quantiles which will be used to determine the output prediction intervals. Therefore, its task is not much more difficult in principle than that of CQR [29], as detecting skewness in the data is relatively easy. However, the additional flexibility of our method can be very advantageous, in terms of the efficiency of the output prediction intervals, if the data are indeed skewed, as it is often the case in real-world applications.
>   Regarding the relevance of our applications, we would like to remind the reviewer that Section 4.3 applies our method to several real data sets which were previously utilized to benchmark earlier conformal prediction methods. In all of these real data sets, which have many features and are fairly complicated, our method significantly outperforms the state-of-the-art alternatives. We feel that this is a fairly compelling demonstration.
>
>
> Questions
>
>  - We would be happy to provide some additional explanations in Section 2.2 to clarify the motivation for our approach. First, we choose to describe our algorithm using a discrete index rather than a continuous-valued variable tau because we want to be concrete, and some discretization is necessary to implement our method in practice. Second, the starting point t-bar is important due to the nestedness constraints. If one did not choose the starting point t-bar correctly, our theoretical oracle results would not hold because the solution must asymptotically become (unconstrained) optimal at the desired nominal level. We understand that these necessities make Section 2.2 a bit technical, but these are not irrelevant details. They are important subtleties that allow our method to simultaneously (a) satisfy desirable theoretical properties, (b) be principled and statistically rigorous, (c) be practical, and (d) perform well on real data. That being said, we do agree that the randomization scheme might make the notation a little too heavy, and so we would be happy to present a simpler non-randomized version of our approach in Section 2.2 and move the randomized one to the supplementary material, as also suggested by reviewer QWwn.
>
>
> Minor
>
>  - Thank you for pointing out the typo on line 188.
>
>  - Thank you for pointing out the typo on line 16 of the supplement.
>
> Justification for score
>
>  - It is true that this method is inspired by previous related work, but we do acknowledge the similarities and give credits for the sources of inspiration. That being said, we are glad the reviewer appreciates that this paper still finds room for improvement in this existing literature, offering something that is novel and practically useful.

---

> > ### Comment · Reviewer_mmq9 · 2021-08-11
> > **Thank you for your comments**
> >
> > Thank you to the authors for their helpful response.
> >
> > - Thank you for clarifying the differences in asymptotic oracle performance between your work and Dist-Split; I missed that oracle defined by loss function in Dist-Split is symmetric. The distinction with that work is now clear, and I understand the desire for predictive intervals rather than sets from your comments to the other reviewers.
> >
> > - By scalable density estimator, I simply meant a single parametric one (perhaps something like Real NVP, but I'm not that familiar) instead of learning and evaluating a large collection of quantile estimators to create the histograms. To clarify, this suggestion is only cursory---I am then unsure of an efficient solution to the optimization in (7). The main "concern/limitation" expressed here is that to actually realize the asymptotic results of 3, the computational complexity for inference becomes quite large. This might be unavoidable in regression while upholding the theoretical desiderata of this work. Out of curiosity, what is the runtime of the current method vs, say, CQR?
> >
> > - I understand that high-dimensional regression is a general challenge. This method requires computing m high-dimensional regression models, as opposed to, say, just two for CQR. It would be great if the authors could further comment on if a larger net error accumulation (in practice, obviously not under the consistency assumptions) could be a problem.
> >
> > Additionally, I do agree with the authors that overall the paper is well cited/situated and is a strong and valuable contribution.

---

> > > ### Author Response · Authors · 2021-08-11
> > > **Follow-up on density estimator and high-dimensional scaling**
> > >
> > > Dear Reviewer mmq9,
> > >
> > > Thank you for your follow-up comments and questions, as well as for the kind assessment.
> > >
> > > We now better understand the original question, thank you for your patience. You are correct that parametric estimators might work well in combination with our method with some data sets, plausibly sometimes even better than the non-parametric ones used in our paper. We did not really explore the use of such parametric approaches because their performance would likely be more heavily dependent on the specifics of the data at hand (we expect they should work better if the model assumptions are well-specified, and worse otherwise). While the choice of estimator is certainly important in applications, we thought it may be better to focus on the novel conformity scores and leave these choices to the practitioners, as they should be made on a case-by-case basis, preferably using prior domain knowledge about the data if that is available.
> > >
> > > While the above argument explains why our paper mostly presents applications of our method with non-parametric estimators, the reason why we focus on multi-quantile regression is that we wanted to make the comparison with CQR as direct and clear as possible. Based on the existing literature, on our empirical experience, and on anecdotal evidence from other researchers, we knew CQR works very well in practice, and so we thought it would make sense to take its underlying quantile regression model as a starting point for our method. In particular, this choice has two advantages. First, it allows us to separate as much as possible the impact of the base regressor from that of the new conformity scores, making the comparison with CQR more informative. Second, it better positions us to achieve state-of-the-art length in the prediction intervals, as we already know deep (or random forest) quantile regression often works relatively well. We did however try to apply our methods in combination with a Bayesian additive regression tree model (this option implemented in our software), which is a semi-parametric approach that does not perform very well on the data considered in this paper, as its modeling assumptions are not well-matched (it makes inappropriate symmetry and homoschedasticity assumptions).
> > >
> > > Regarding the computational cost of the deep multi-quantile regressor employed by our method, that is actually comparable to that of the bi-quantile regression model used by CQR. The reason is that the numbers of parameters and the architecture of the neural architectures are the same in both cases, the only difference is that our model has more than two outputs (all sharing the same input and inner layers). Therefore, the computational cost and runtime of training the network are approximately the same. Intuitively, this can be understood as thinking of the neural network as learning an approximate representation of the conditional distribution of Y|X, regardless of how many different quantiles are explicitly estimated. Of course, that is not to say that estimating many quantiles is as easy as estimating only two, but most of the additional statistical difficulty would come from estimating extremely large or small quantiles, not the intermediate ones. Precisely to avoid this problem, our model does not attempt to estimate extremely large or small quantiles (below 1% or above 99%); instead, the tails are smoothed using a simple prior (see Section S1.1).
> > >
> > > The above paragraph also begins to answer the next question about the statistical challenge of high-dimensional regression. If the multi-quantile regression model is implemented with the standard approach suggested in our paper (e.g., with the deep neural network), we don't think there is a problem of error accumulation across quantiles. Each quantile is estimated by a separate output node based on the same "pinball" loss function used by the simpler model employed by CQR. Indeed, we expect the 5% and 95% quantiles estimated by our multi-quantile model should be quite similar to those one would obtain by training the same neural network to estimate solely the 5% and 95% quantiles, although it is true we did not explore this intuition in much depth. Instead, the main additional difficulty in multi-quantile regression is the estimation of extremely small or large quantiles, but again we refrained from going beyond 1% or 99% precisely for this reason, preferring a smoothing approach instead. The reasons why we did not dive deeper into the deep learning tradeoffs of estimating multiple quantiles are that (a) the moderate range of quantiles we estimate (between 1% and 99%) is more than sufficient for our purposes, and (b) multi-quantile regression models are already quite standard, see for example "Learning Multiple Quantiles With Neural Networks" by Moon et al 2021, or "Quantile Regression Forests" by Meinshausen 2006).
> > >
> > > We hope these comments answers your questions satisfactorily. Thank you for giving us the opportunity to clarify these important points, which should certainly receive a little more attention in the revised version of the manuscript.

---

> > > > ### Comment · Reviewer_mmq9 · 2021-08-11
> > > > **Thank you**
> > > >
> > > > Thanks, this explanation was quite helpful.

---

### Official Review · Reviewer_U8cY · 2021-07-16

**Rating:** 7
**Confidence:** 4

**Summary:**

This paper considers constructing prediction set using histogram estimation of conditional density. The results and methods are new and interesting.

**Ethics Review Area:**

["I don’t know"]

**Limitations And Societal Impact:**

The term "conformal" needs to be further explained. As to common readers, the current approach is a "plug-in" method since unknown density is replaced by its estimator. Authors should explain the explicit reason of using "conformal" to name this procedure.

Second, can $Y_i$ be multi-dimensional?

Third, can $f$ be estimated using general kernel density estimator?




**Main Review:**

Prediction set based on estimated density seems a natural and fundamental problem. The current paper provides a novel method and theory regarding this problem. Paper writing is clear and results are significant.

**Time Spent Reviewing:**

2

---

> ### Author Response · Authors · 2021-08-09
> **Response to reviewer U8cY**
>
> We are grateful to reviewer U8cY for the thoughtful review and the encouraging feedback.
>
> Our proposed method is a conformal inference method in the sense that its prediction sets are calibrated by analyzing the distribution of suitable conformity scores evaluated on hold-out data, independent of the training data, in such a way as to guarantee marginal coverage. We thought most readers would at this point have at least some familiarity with conformal inference, although this reviewer's comment suggests that perhaps this may not be the case. Currently, Section 1.1 mentions that this is a conformal inference method and points to the relevant fundamental literature for the benefit of readers that are not yet familiar with conformal inference. Unfortunately, it would not be compatible with the space limitation to make this paper fully self-contained for the readers who do not know what conformal inference is. However, we would be happy to spend a few more introductory words in the introduction if the reviewers deem it necessary.
>
> Regarding a multi-dimensional Y, this is a very good question. As currently presented, our method is designed to predict a univariate response, and we feel that relaxing this assumption would make the notation in the paper too heavy (other reviewers have already mentioned that this is not a light paper). However, it is true that it would be relatively simple, from a conceptual standpoint, to extend our approach to deal with a multi-dimensional Y. We would be happy to mention this interesting opportunity for future work if given the opportunity to do a minor revision.
>
> This is also a good comment. Yes, as hinted by the reviewer, any density estimator could be utilized to estimate the conditional distribution. Indeed, we mentioned this intrinsic flexibility in Section 2.1, along with a non-exhaustive list of possible alternatives to quantile regression. However, we would be happy to emphasize this flexibility even more if the reviewers deemed it necessary.

---

### Official Review · Reviewer_P36z · 2021-07-17

**Rating:** 8
**Confidence:** 3

**Summary:**

This paper introduces a new method to construct conformal prediction intervals based on histograms of the conditional distribution of an outcome variable. Given a histogram of the conditional distribution of the outcome variable the method finds the shortest interval whose associated mass is no less than the desired coverage probability of the prediction interval. This yields intervals that automatically adapt to the skewness of the data. The intervals have provably correct marginal coverage in finite samples and correct conditional coverage and optimal length in large samples. A simulation study and numerical experiments on several benchmark data sets corroborate the theoretical results and demonstrate the advantage over several competing methods.


**Limitations And Societal Impact:**

Yes, the authors have adequately addressed the limitation of their assumptions and the lack of control of lower and upper miscoverage rates.

**Main Review:**

This a well-written and clearly organized paper that makes an interesting contribution to conformal prediction. The paper addresses the important problem of how to construct conformal confidence sets that have finite-sample marginal coverage and large-sample conditional coverage while being as narrow as possible. Typically, adaptivity and optimality of (conformal) confidence sets are only argued heuristically (by judiciously choosing tuning parameters, e.g. Lei and Wasserman, JRSSB 2014) and are often excluded from the theoretical analysis. The novelty of the proposed method is that adaptivity and optimality are guaranteed.

To me the strength of the paper lies in the simple and intuitive idea of the proposed method (setting aside the somewhat tedious actual implementation). I believe that the basic idea of directly minimizing the length of the prediction interval subject to a constraint on the coverage probability can be extended and adapted to related problems. Thus, the paper does not simply propose a solution to an important problem, but also provides food for thought (see below).

An obvious weakness of the paper are the strong assumptions for the theoretical analysis. I am fine with Assumptions 1-3, but I would have preferred if the authors had done without Assumption 4 and 5 on the unimodality of the true and estimated conditional density of the outcome variable. At the very least, the authors could have included empirical results from an outcome variable with non-unimodal conditional distribution.

I wonder whether the authors have thought about the following extension of their approach:  Why "only" consider prediction intervals of shortest length and not more general prediction regions with smallest Lebesgue measure? It seems to me that opt. problem (7) can be easily modified in such a way that even non-unimodal conditional distributions of the outcome variable can be handled. Consider solving

$\mathcal{S}(x, \pi, S^-, S^+, \tau) :=  \arg\min_{M\subseteq \\{1,\ldots, m\\}} \left\\{ |M| :  \sum_{j \in M} \pi_j(x) \geq \tau, S^- \subseteq M \subseteq S^+ \right\\} \quad{}\quad{} (*)$

If the histogram of the conditional density of the outcome variable is unimodal, then $\mathcal{S}(x, \pi, S^-, S^+, \tau)$ is an interval. Otherwise it is a collection of convex sets. Therefore, (\*) seems to be even more adaptive than the original opt. problem (7).

**Time Spent Reviewing:**

2

---

> ### Author Response · Authors · 2021-08-09
> **Response to reviewer P36z**
>
> We are grateful to reviewer P36z for the thoughtful review and the encouraging feedback. We feel flattered by such a high score.
>
> Assumptions for theoretical analysis.
>
> This is a very good comment, and we thank the reviewer for the opportunity to expand even more on the discussion of our assumptions. It is true that our assumptions are quite strong, (1-3 in particular) but they are useful. First, the main purpose of our theoretical analysis is not to reassure the practitioner that our method achieves the oracle intervals on a particular data set. Obviously, nobody can promise something like that. Instead, the main purpose of our theoretical analysis is to highlight the conceptual advantage of our method relative to the existing alternatives for computing conformal prediction intervals, and to give some deeper insight to our readers as to how our method works and under which conditions it can be expected to perform best. Second, the assumptions mostly involve the estimation of the conditional distribution, which is performed by a black box that we deliberately want to keep as non-parametric and flexible as possible, and not the actual procedure by which we calibrate the output prediction intervals, which is the heart of our method.
>
> It is true that we could have done without assumption 5 but, as stated in section 3, this is not a very strong assumption and so not much is lost by retaining it. Indeed, assumption 5 is not very strong because it is almost implied (loosely speaking) by assumptions 2 and 4. Further, assumption 5 could actually be easily satisfied in practice by suitably regularizing the black-box estimator of the conditional distribution. Regarding assumption 4, this indeed places some burden on the data-generating distribution, but we do not think it would be very meaningful to relax it. In particular, we think unimodality is essential within this theoretical analysis, because our method is specifically designed to compute prediction intervals, not general non-convex sets. Therefore, the most significant sanity check is to verify whether it can indeed recover the optimal prediction intervals, under suitable asymptotic conditions, in settings where prediction intervals are most meaningful. If it is known a priori that the true data-generating distribution is not uni-modal, perhaps one should use a different method to compute non-convex prediction sets, such as the ones we cite from [20].
>
> The next comment about relaxing the constraints that our predictions be intervals is closely related to the last part of our answer above.  If it is known a priori that the true data-generating distribution is not uni-modal, one should apply the methods from [20], which are very closely related to this reviewer's proposal. The reason why our paper focuses on prediction intervals rather than arbitrary sets is that intervals are (a) naturally more interpretable in many applications, and (2) less prone to overfitting. Please see our answer to a closely related comment by reviewers prSn and ucSx, where we further motivate our interest in prediction intervals.

---

### Official Review · Reviewer_prSn · 2021-07-24

**Rating:** 7
**Confidence:** 5

**Summary:**

This paper proposes a new nonconformity-score based on conditional density estimates P(Y|X). The main idea is to approximate the optimal prediction interval based on the conditional density estimate. The proposed non-conformity score is experimentally assessed using the conformalization technique of split conformal. The overall method provably achieves marginal coverage, performs comparably to other methods for conditional coverage, and achieves the shortest prediction intervals on all datasets.

**Limitations And Societal Impact:**

Yes

**Main Review:**

My overall opinion of this paper is quite positive. The method makes sense and the experiments indicate a significant improvement on the efficiency/length of the prediction intervals. Thus I think the paper is a useful contribution to the ML community. In the following 'sections' I elaborate further on things that I liked about the paper, along with some suggestions for further improvements.

However, I have one methodological question that is almost 'begging to be answered'. The optimal prediction **set** given P(Y|X) is simply to take the highest predictions until 1-\alpha coverage is reached (Appendix F [2]). In the case of viewing it from a 'histogram' perspective, one would take the largest histogram bars until 1-\alpha is reached. Perhaps considering the optimal prediction **interval** is more interpretable in some applications, as the authors do. However, in some applications, I think sets could also be ok? Thus, I think it makes sense to also explore if taking prediction sets helps reduce the widths. If the authors found in experiments that it does not help, could the authors briefly comment on this? Also, what if one takes the convex-hull interval of the PS to make it a PI?

On a related note, lines 43-46 do not answer the following question. P(Y|X) is not known, but after an estimate of P(Y|X) is computed, why is it discretized into bins? As far as I can tell, the P(Y|X) models that the authors use essentially have real-valued output. Why not work with the full real-valued estimate and define nested sets for them?

*Writing and clarity:*
The introduction was clean and easy-to-follow. By the end of page 2, I already had a broad idea of what to expect from the rest of the paper.
I have a significant concern regarding the density of notation and ideas introduced in Sections 2.2 and 2.3. In lines 165–167, the authors say: “In particular, we analyze a slightly modified version of Algorithm 1 in which there is no randomization; this is theoretically more amenable and equivalent in spirit, although it may yield wider intervals in finite samples.” I fully agree. In light of this, is there a reason the authors chose to describe the full randomization strategy along with the main conformal algorithm in Section 2.2 (rather than in a future section or Appendix)? In my opinion, Section 2.2 has way too much notation and many nitty-gritties of conformal inference are introduced very densely…. I think this could make the current description of the method *quite* difficult to grasp for someone unfamiliar with conformal prediction.

*Originality:*
The main idea of using P(Y|X) estimates to produce efficient conformal sets has appeared in different forms, and is not striking. In this sense, the initial prediction sets that the authors describe (Sec 1.2) are natural. However, they pose the issue of not being nested sets making it tricky to apply conformal prediction. I really liked the way the authors translated this setup into a nested problem by first fixing a single set (eq. (9)), and then considering supersets and subsets of this initial set (eq. (10)). The experimental section verifies that this is a useful description of nested sets for producing short prediction intervals.

*Terminology:*
In the nonparametrics statistics literature, histogram classification and histogram regression have been used to refer to methods that partition the feature space X into a number of bins, and compute a single prediction for each bin [1, and many others]. To the best of my understanding, there is very little intersection between the ideas proposed by the present authors, and the histogram regression literature. Due to this, I strongly suggest changing the title of the paper and the name of the proposed method. Something as simple as “Conformal inference using conditional histograms” avoids the confusion, and avoids overstepping on another interesting body of work.

Minor comments:
- Space permitting, it would be great if Assumptions 1–5 are latex enumerated.
- It may be useful to demonstrate that the prediction interval based on the unconformalized conditional density estimate (based on Sec 1.2) does not have valid coverage.

References:

[1] https://www.jstor.org/stable/2242583

[2] https://arxiv.org/pdf/1910.10562.pdf

**Time Spent Reviewing:**

6

---

> ### Author Response · Authors · 2021-08-09
> **Response to reviewer prSn**
>
> We are grateful to reviewer prSn for the thoughtful review and the encouraging feedback.
>
> ### Why prediction intervals
>
> Thank you for giving us the chance to motivate further the relevance of our method for computing prediction *intervals*, as opposed to prediction *sets*. This is a crucial point which deserves even more discussion than we were able to allocate within the tight page constraints.
>
> While we certainly do not claim they are universally preferable, prediction intervals are often naturally much more interpretable than arbitrary regions. Further, reporting a non-convex prediction region conveys a level of statistical confidence that may be hard to justify. For example, if we were to report to a physician that the likely blood pressure or a patient with certain characteristics will, at some point in the future, be within the interval [120,129] mm Hg, we are likely to be of help. However, a statistician who reports to the physician that the patient's future blood pressure is likely to be within the following region, ( [120, 120.012] U [120.015, 120.05] U [121, 122.7] U [123.1, 127.2] U [127.8, 129] ) mm Hg, would risk being taken less seriously. Indeed, it would not be clear (a) whether the multi-modality is significant or a spurious consequence of overfitting, and (b) how the physician would act upon this prediction any differently than if it had been [120,129] mm Hg.
> As thoughtfully suggested by this reviewer, the advantage of prediction intervals over sets could be emphasized more explicitly. We would be happy to do so if given the opportunity to do a minor revision.
>
> Finally, the method suggested by the reviewer for constructing possibly non-convex prediction sets is very closely related to the work of [20], which we already cite. It is however worth emphasizing that much of the technical difficulty, and innovation, or our work arises precisely from the added requirement that the prediction sets be (the shortest possible) intervals.  Of course, one can always transform a one-dimensional non-convex predictive region into an interval by taking its convex hull. However, if we know from the beginning that we wish to report an interval, why take such an indirect route? Certainly in that case it would seem more intuitive and statistically sound to apply a method that is explicitly designed to find the shortest predictive intervals with valid coverage, rather than to resort to heuristic patchworks of different approaches. Please note that a similar comment was also raised by reviewer ucSx.
>
> ### Discretization
>
> The first reason why we discretize the problem is that we want the proposed method to be practically implementable, and it is not clear how one would solve the necessary optimization problems without some form of discretization. The second reason is that it is not realistic to expect one can learn the conditional distribution of Y|X to an arbitrary degree of precision. Instead, what we do in practice is to fit a multi-quantile regression model, which gives us a discretized approximation of the true conditional distribution that can, loosely speaking, already be thought of as a histogram. The only meaningful difference between the multi-quantile regression model estimated by the machine-learning black-box model and the conditional histogram based on which our conformity scores are defined is the choice of bins. Although it would not be necessary in principle, we prefer to transform the multi-quantile regression model into a histogram representation with pre-defined bins because this significantly simplifies the exposition, and the implementation, of our method, allowing us to fully decouple the black-box estimation of the conditional distribution from the computation of the conformity scores. Of course, part of the appeal of this approach is that it highlights how our method is not actually tied to any multi-quantile regression model, but it can be easily implemented also with other models, as mentioned in the paper.
>
> ### Writing and clarity
>  We agree that the exposition may benefit if we defer the randomization to the appendix, as that will simplify the notation without much loss of essential content. We will do so if given the opportunity to do a minor revision. This was also suggested by reviewer QWwn.
>
> ### Terminology
>
>  Thank you for suggesting the alternative title "Conformal inference using conditional histograms". We agree that this suggestion is clearer, so we would be happy to change the title accordingly.
>
> ### Minor comments
>
>  - We will be happy to enumerate in latex Assumptions 1–5 in a revised manuscript, if space allows it.
>  - It is a good idea to show explicitly that the unconformalized conditional density estimate does not have valid coverage. We will do so if given the opportunity to do a minor revision.

---

### Decision · Program_Chairs · 2021-09-27

**Decision:**

Accept (Spotlight)

**Comment:**

The paper presents an original contribution for conformal prediction with optimized bandwidth. The paper constructs conformal confidence sets that have finite-sample marginal coverage and large-sample conditional coverage while being as narrow as possible. This, I believe, will be a valuable addition to the current conformal prediction / regression problems, as confirmed by reviewers.